# Genomic instability caused by Arp2/3 complex inactivation results in micronucleus biogenesis and cellular senescence

Elena L. Haarer[1,2], Corey J. Theodore[1,2], Shirley Guo[1,2], Ryan B. Frier[1,2], Kenneth G. Campellone[1,2,3]*

1 Department of Molecular and Cell Biology; University of Connecticut, Storrs, Connecticut, United States of America, 2 Institute for Systems Genomics; University of Connecticut, Storrs, Connecticut, United States of America, 3 Center on Aging, UConn Health; University of Connecticut, Storrs, Connecticut, United States of America

* kenneth.campellone@uconn.edu

**Data Availability Statement:** All relevant data are within the manuscript and its Supporting Information files.

## Abstract

The Arp2/3 complex is an actin nucleator with well-characterized activities in cell morphogenesis and movement, but its roles in nuclear processes are relatively understudied. We investigated how the Arp2/3 complex affects genomic integrity and cell cycle progression using mouse fibroblasts containing an inducible knockout (iKO) of the ArpC2 subunit. We show that permanent Arp2/3 complex ablation results in DNA damage, the formation of cytosolic micronuclei, and cellular senescence. Micronuclei arise in ArpC2 iKO cells due to chromatin segregation defects during mitosis and premature mitotic exits. Such phenotypes are explained by the presence of damaged DNA fragments that fail to attach to the mitotic spindle, abnormalities in actin assembly during metaphase, and asymmetric microtubule architecture during anaphase. In the nuclei of Arp2/3-depleted cells, the tumor suppressor p53 is activated and the cell cycle inhibitor *Cdkn1a*/p21 mediates a G1 arrest. In the cytosol, micronuclei are recognized by the DNA sensor cGAS, which is important for stimulating a STING- and IRF3-associated interferon response. These studies establish functional requirements for the mammalian Arp2/3 complex in mitotic spindle organization and genome stability. They also expand our understanding of the mechanisms leading to senescence and suggest that cytoskeletal dysfunction is an underlying factor in biological aging.

## Author summary

The actin cytoskeleton consists of protein polymers that assemble and disassemble to control the organization, shape, and movement of cells. However, relatively little is understood about how the actin cytoskeleton affects genome maintenance, cell multiplication, and biological aging. In this study, we show that knocking out the Arp2/3 complex–a core component of the actin assembly machinery in mammalian cells–causes DNA damage, genomic instability, mitotic chromosome partitioning defects, and a permanent cell proliferation arrest called senescence. Since senescent cells are major contributors to both

**Funding:** KGC was supported by National Institutes of Health grants R01-GM107441 and K02-AG050774 (www.nih.gov). The funders had no role in study design, data collection and analysis, decision to publish, or preparation of the manuscript.

**Competing interests:** The authors have declared that no competing interests exist.

age-associated diseases and tumor suppression, our findings open new avenues of investigation into how natural or experimental alterations of cytoskeletal proteins impact the process of aging and the regulation of cancer.

## Introduction

The actin cytoskeleton consists of dynamic protein polymers that have well-known functions in cell morphogenesis and motility. Globular (G-) actin monomers are present in the cytoplasm and nucleus, and their polymerization into filamentous (F-) actin is driven by proteins called nucleators [1]. These include actin monomer-oligomerizing proteins, Formin-family nucleation/elongation proteins, and the Arp2/3 complex–a heteroheptameric actin assembly factor that binds to the sides of existing filaments and nucleates new filaments to create branched networks [2]. The Arp2/3 complex is highly conserved across almost all eukaryotes [3,4] and is required for viability in such organisms; inactivation of genes encoding its subunits prevents growth of *S.cerevisiae* [5,6] and *D.discoideum* [7] and is embryonic lethal in animals including *D.melanogaster* [8,9], *C.elegans* [10,11], and *M.musculus* [12–14]. However, the cellular basis underlying the essential nature of the Arp2/3 complex is not well understood.

Many processes that involve plasma membrane dynamics, especially cell adhesion and migration, rely on actin networks assembled by the Arp2/3 complex [15]. In fact, conditional knockouts in mice indicate that the complex is crucial for maintaining normal tissue architecture, promoting changes in cell shape, and powering cell migration during development [14,16–20]. These *in vivo* results are consistent with molecular and cellular studies of Arp2/3-mediated actin assembly using *in vitro* systems [21], dominant negative regulatory proteins [22], transient RNAi-mediated knockdowns [23], and pharmacological inhibitors of the complex [24,25].

In contrast to the well-characterized roles of the Arp2/3 complex in protrusion and motility, its functions in nuclear processes are only beginning to emerge. During interphase, all 3 classes of actin nucleators promote nuclear actin filament assembly in response to DNA damaging agents [26–28]. In *Drosophila* and mammalian cells, Arp2/3-mediated actin polymerization is crucial for repositioning damaged heterochromatin to the nuclear periphery, which enables subsequent DNA repair activities [27,28]. Moreover, depletion of the Arp2/3 complex using RNAi in *Drosophila* larvae leads to chromosomal abnormalities and genomic instability [27]. Additional studies in human cells exposed to DNA damaging agents indicate that depletion of the Arp2/3 complex also causes defects in pro-apoptotic signaling [29].

Apart from their functions in chromatin-associated processes during interphase, actin and its nucleators, especially Formins and the Arp2/3 complex, are increasingly being found to support proper chromosome movements during meiosis and mitosis. In starfish oocytes, after nuclear envelope breakdown, several types of F-actin structures promote chromosome transport and coordinate capture by microtubules [30,31]. Studies in mouse oocytes further show that actin filaments permeate the meiotic microtubule spindles and facilitate proper chromosome congression [32,33]. Chemical inhibition of actin dynamics or genetic inactivation of Formin-2 prevents proper formation of kinetochore microtubules and leads to chromosome alignment and segregation errors [33,34]. Similarly, during mitosis, several actin structures have been shown to interact with and possibly guide microtubule spindle components. Actin filaments that run between the microtubule spindle poles and F-actin fingers that project from the cell cortex into the spindle have been identified in *Xenopus* epithelial cells [35]. Centrosomes, which serve as major microtubule nucleation and organizing centers, are also sites of

actin assembly [36]. The Arp2/3 complex localizes to centrosomes in multiple mammalian cell types, and pharmacological inhibition of Arp2/3 results in decreased centrosomal actin levels and impaired mitotic spindle formation [37–39]. Thus, disruption of either actin or Arp2/3 function during meiosis or mitosis can lead to defects in chromosome dynamics, highlighting the actin cytoskeleton as a key player in maintaining genomic integrity during nuclear division.

Although the effects of transient Arp2/3 depletion or inhibition on chromatin repair are now evident, and several aberrations in chromosome movement have been characterized, the impact of total and permanent Arp2/3 ablation on these processes has not been established. The development of several cellular systems for studying long-term Arp2/3 depletion or deletion has allowed more clear-cut assessments of the requirements for the Arp2/3 complex in a given cellular process [14,40–45]. For example, these models have already provided fundamental insights into the function of the Arp2/3 complex in cell migration. Studies using mouse embryonic fibroblasts (MEFs) expressing shRNAs targeting the ArpC2 and Arp2 subunits [40], embryonic stem cell-derived fibroblasts lacking the ArpC3 subunit [14], fibroblasts harboring tamoxifen-inducible knockouts of the ArpC2 or Arp3 subunits [42,43], and human neutrophil-like cells depleted of Arp2 [44,45] indicate that the Arp2/3 complex is crucial for cell polarization, lamellipodia formation, and/or directional migration in several environmental contexts.

To determine the outcome of Arp2/3 complex ablation on chromatin-associated processes related to cell viability and multiplication, we employed the inducible ArpC2 knockout cell model [42]. Our findings connect Arp2/3 complex functions in maintaining genomic integrity during interphase and mitosis in normal cells to the biogenesis of micronuclei and induction of cellular senescence pathways when Arp2/3 is inactivated.

## Results

### ArpC2 iKO cells undergo an abrupt proliferation arrest and morphological enlargement

Given that the Arp2/3 complex is required for viability in many eukaryotic organisms, we sought to better understand its essential nature–apart from its well-recognized roles in adhesion and motility–in mammalian cells. The Arp2/3 complex is composed of seven subunits: two Actin-related proteins (Arp2 and Arp3) and five Complex subunits (ArpC1-C5), with ArpC2 and ArpC4 forming a structural core and multiple isoforms of Arp3, ArpC1, and ArpC5 providing peripheral diversity [4,46]. Previous work indicates that MEFs subjected to RNAi-mediated depletion of the ArpC2 and Arp2 subunits are viable and remain culturable when generated in a genetic background lacking the *p16Ink4a/Arf* tumor suppressors [40]. More recently, to circumvent problems with knockdown instability, a conditional knockout model was created using *p16Ink4a/Arf $^{-/-}$* mouse tail fibroblasts (MTFs) harboring a floxed *Arpc2* allele and engineered to induce CreER recombinase activity upon treatment with 4-hydroxytamoxifen (4-OHT) [42]. Since the latter system is inducible, causes a permanent loss of the critical ArpC2 subunit, and leads to degradation of other members of the complex, we adopted this Arp2/3 complex null cellular model for our studies. In all experiments, parental MTFs carrying the conditional *Arpc2* allele were exposed to DMSO to maintain a control (Flox) cell population or to 4-OHT to generate ArpC2 induced knockout (iKO) cells.

For initially assessing the kinetics of Arp2/3 complex depletion, DMSO- and 4-OHT-treated cells were collected at various timepoints, lysed, and immunoblotted with antibodies to the ArpC2 and Arp3 subunits plus antibodies to GAPDH and tubulin as controls (Fig 1A). After 1 day in 4-OHT, ArpC2 protein levels were reduced by nearly half compared to DMSO-

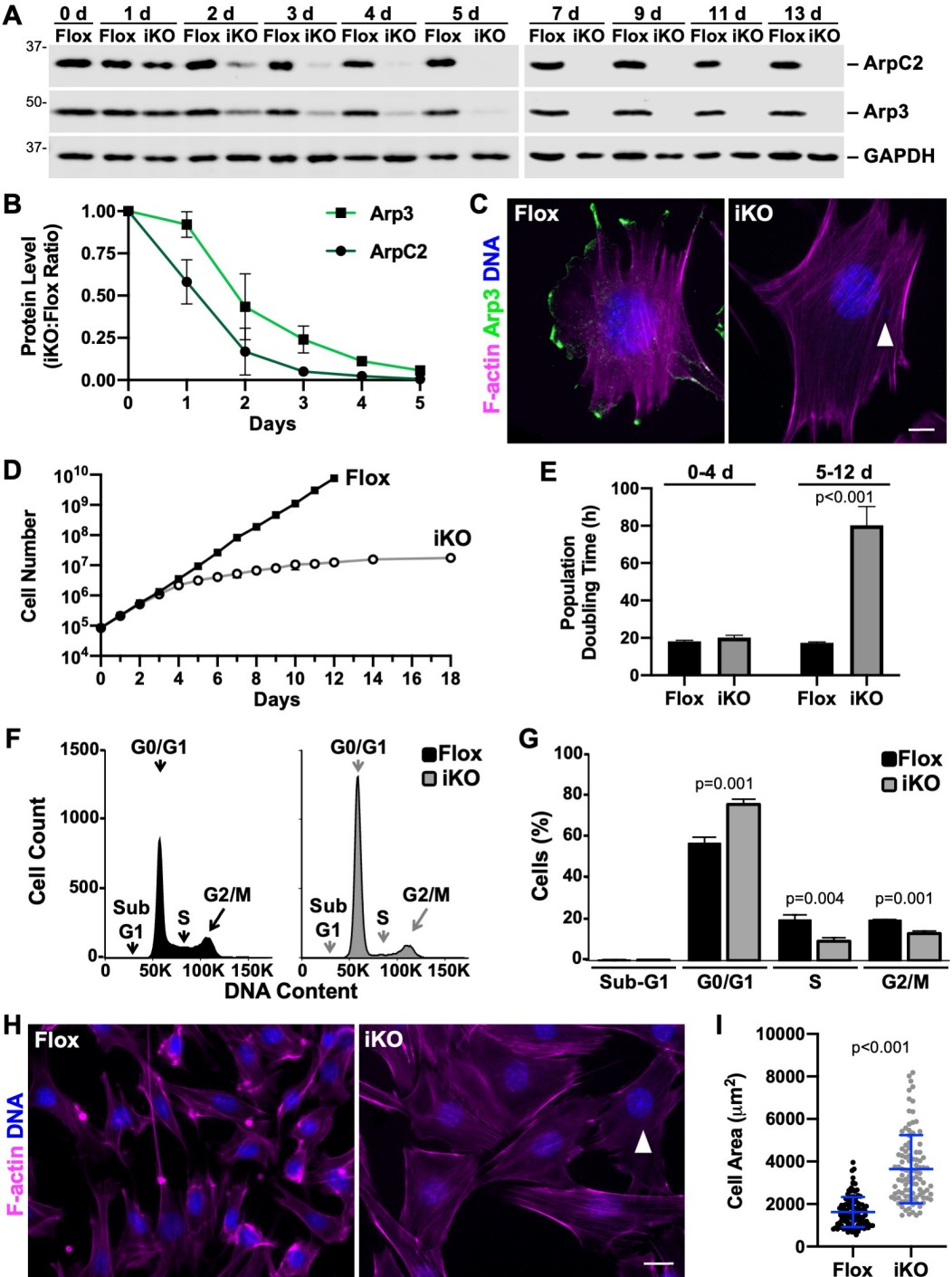

**Fig 1. ArpC2 iKO cells undergo a proliferation arrest and enlargement. (A)** Mouse fibroblasts were treated with DMSO (Flox) or 4-OHT (iKO) for 0-6d and collected at 0-13d. Samples were lysed, subjected to SDS-PAGE, and immunoblotted with antibodies to ArpC2, Arp3, and GAPDH. **(B)** ArpC2 and Arp3 band intensities were normalized to GAPDH or tubulin band intensities and plotted as the iKO:Flox ratio. Each dot represents the mean ratio ±SD from n = 2–3 experiments. **(C)** Flox and iKO cells were fixed at 7d and stained with phalloidin (F-actin; magenta), an Arp3 antibody (green), and DAPI (DNA; blue). The arrowhead highlights a micronucleus. Scale bar, 10μm. **(D)** Flox and iKO cell titers were quantified at 0-18d. Each point represents the mean ±SD from n = 2 Flox and n = 2–3 iKO experiments, except for the 14d and 18d timepoints, which were from a representative iKO sample that did not exhibit outgrowth of colonies expressing ArpC2. **(E)** Flox and iKO population doubling times were quantified daily from 0-4d and 5-12d. For each time range, the bar represents the mean doubling time ±SD from n = 3 experiments. **(F)** Flox and iKO cells were fixed at 9d, stained with propidium iodide,

and analyzed by flow cytometry. 20,000 events were examined for each cell type. **(G)** The % of cells in each phase of the cell cycle was quantified. Each bar represents the mean % ±SD from n = 3 experiments. The relevance of iKO cells with 4n content (i.e., appearing to be in G2) are described in live imaging experiments in Fig 5. **(H)** Flox and iKO cells were fixed at 7d and stained with phalloidin and DAPI. The arrowhead highlights a micronucleus. Scale bar, 25μm. **(I)** Flox and iKO cells were outlined in ImageJ and their areas quantified. Each dot represents an individual cell and the blue lines represent the mean area ±SD from analyses of n = 100 cells.

treated Flox cells (Fig 1A and 1B). By 2 days, ArpC2 expression was diminished by approximately 80%, and a reduction in Arp3 levels became noticeable (Fig 1A and 1B). The amounts of both subunits continued to steadily decline in the iKO cells over time until they were absent following 5 days in 4-OHT (Fig 1A and 1B). DMSO and 4-OHT were removed from culture media after 6 days, but ArpC2 and Arp3 remained undetectable in the iKO population out to 13 days (Fig 1A). To independently confirm the loss of the Arp2/3 complex by fluorescence microscopy, Flox and iKO cells were stained with an antibody to label Arp3 and with fluorescent phalloidin to visualize F-actin. As expected [42], Flox cells exhibited prominent Arp3 staining within F-actin-rich peripheral membrane ruffles, whereas Arp3 staining and ruffles were both missing from the iKO cells (Fig 1C). Collectively, these results confirm that in this cellular context, the Arp2/3 complex knockout is rapid, complete, and stable over time.

To determine the impact of abolishing Arp2/3 complex expression on cell proliferation, we next quantified cell titers on a daily basis following the addition of DMSO or 4-OHT. For the first 3 days, Flox and iKO cells multiplied at identical rates, but by 4 days, the growth characteristics of Flox and iKO cultures began to diverge, and at 5 days, the iKO samples were proliferating at a clearly slower pace (Fig 1D). After approximately 10–12 days, virtually all iKO cells stopped dividing (Fig 1D). To quantify the differences in cell multiplication rates, we calculated the population doubling times in the 0–4 and 5–12 day time periods following DMSO or 4-OHT exposure. While the doubling times were similar for Flox and iKO populations (18h vs. 20h) during the first interval, the iKO doubling times quadrupled to >80h in the 5–12 day range (Fig 1E). Moreover, cell counts in iKO samples remained unchanged for an additional week (Fig 1D), except in instances where colonies of 4-OHT escapees or suppressor mutants emerged (S1 Fig). Thus, following loss of the Arp2/3 complex, MTF cells undergo an abrupt and stable proliferation arrest.

To better understand this arrest, we performed fluorescence activated cell sorting (FACS) analyses of the DNA content of Flox and iKO cells. Both Flox and iKO samples had high viability and contained few apoptotic events, as <0.3% of cells in each population had sub-G1 DNA content (Fig 1F and 1G). The lack of iKO cell death was expected, based on the importance of the Arp2/3 complex in intrinsic apoptosis [29]. Further cell cycle comparisons revealed that the iKO population contained significantly more cells in G0/G1 and significantly fewer in S and G2/M at 9 days (Fig 1F and 1G). A similar shift to G0/G1 in the iKO population was observed even earlier, at 6 days (S1 Fig). These results indicate that most iKO cells harbor 2n DNA content and arrest in G1.

When examining the different growth characteristics of the Flox and iKO cultures, it also became apparent that the two cell types had distinct morphologies and sizes. Fluorescent phalloidin staining, in addition to revealing a lack of F-actin-rich ruffles (Fig 1C), demonstrated that iKO cells were usually flatter and larger than Flox cells (Fig 1H), similar to results observed in cells containing an induced knockout of the Arp3 subunit [43]. Quantification of Flox and ArpC2 iKO cell areas showed that by 7 days the iKO cells became, on average, about twice as large as Flox cells (Fig 1I). Again consistent with a requirement for the Arp2/3 complex in intrinsic apoptosis [29], apoptotic cell morphologies were not observed in the iKO cell

population (Fig 1H). So in addition to losing their ability to multiply, Arp2/3-deficient cells display significant increases in their size.

## ArpC2 iKO cells exhibit the canonical nuclear and cytoplasmic features of senescence

A loss of proliferative capacity and an increase in adherent cell area are common characteristics of senescent cells. Cellular senescence refers to a permanent state of replicative arrest [47,48], and is reflected in several additional physiological changes, including increased production of pro-inflammatory proteins, a response known as the senescence-associated secretory phenotype, or SASP [49,50]. Notably, previous global gene expression profiling using the ArpC2/Arp2-depleted MEF model revealed that several SASP genes were up-regulated [41], suggesting a link between the loss of Arp2/3 function and this aspect of cellular senescence. To investigate whether the ArpC2 iKO cells also displayed this senescence feature, we performed RT-PCR (Fig 2A) and RT-qPCR (Fig 2B) to compare transcript levels for Interleukin-6 (Il-6), a pro-inflammatory cytokine consistently present in the SASPs derived from senescent cells of diverse origins [51,52]. In agreement with findings from the ArpC2/Arp2 RNAi MEF studies, *Il-6* expression was greater in iKO cells than in Flox cells at 3, 6, and 9 days after the onset of 4-OHT treatment (Fig 2A). RT-qPCR revealed that *Il-6* transcript levels were nearly 4-fold higher in the iKO compared to Flox cells at 9 days (Fig 2B), indicating that a permanent loss of the Arp2/3 complex induces the production of this key SASP component.

Other cytokines that may be upregulated in some senescent cell types include Interleukin-1-beta (Il-1β) and Interferon-beta (Ifn-β) [51–53]. While immunoblotting suggested that iKO cells do not overproduce Il-1β (S2 Fig), RT-PCR indicated that iKO cells do contain significantly elevated levels of *Ifn-β* transcript (Fig 2C and 2D). Therefore, Arp2/3 complex ablation, in addition to increasing *Il-6* expression, apparently leads to an interferon response.

Cellular senescence is also frequently associated with changes in nuclear structure, including decreased levels of Lamin B1, a cytoskeletal protein of the nuclear lamina [54]. To determine if Lamin B1 abundance was altered by the deletion of ArpC2, we immunoblotted Flox and iKO cells for Lamin B1 and found that Lamin B1 protein levels were 3-fold lower in iKO cell populations (Fig 2E and 2F). To confirm the reduction in Lamin B1 expression, Flox and iKO cells were also subjected to immunofluorescence microscopy. In Flox cells, nuclear Lamin B1 staining was consistently bright (Fig 2G and 2H) and mostly nucleoplasmic (Fig 2G, magnifications). Contrastingly, in iKO cells, nuclear Lamin B1 levels were visibly lower (Fig 2G). In agreement with the immunoblotting data, quantification of nuclear fluorescence demonstrated that, on average, Lamin B1 was nearly 3-fold less intense in iKO cells (Fig 2H).

In addition to the production of a transcriptional cytokine response and a decrease in nuclear Lamin B1 levels, senescent cells typically exhibit a cytoplasmic senescence-associated β-galactosidase (SA-βgal) activity at pH 6 [55]. This is the most widely accepted biomarker of senescence. SA-βgal staining in Flox and iKO cells from 0–22 days revealed that by 7 days, and at later timepoints, the number of SA-βgal-positive cells was significantly higher in iKO than Flox populations (Fig 3A and 3B). In many instances, SA-βgal activity is thought to reflect an increase in lysosomal content [56,57]. So to assess lysosomal abundance, we treated Flox and iKO cells with LysoTracker, a fluorescent probe that labels acidic intracellular structures, and examined the cells microscopically. LysoTracker intensely stained discrete circular puncta resembling lysosomes in Flox cells, but broadly stained large portions of the cytoplasm in iKO cells (Fig 3C). Quantification of LysoTracker fluorescence at day 9 revealed that more than 80% of iKO cells versus 5% of Flox cells showed the more diffuse staining pattern (Fig 3D). Furthermore, >10% of the area within iKO cells stained positive for LysoTracker compared

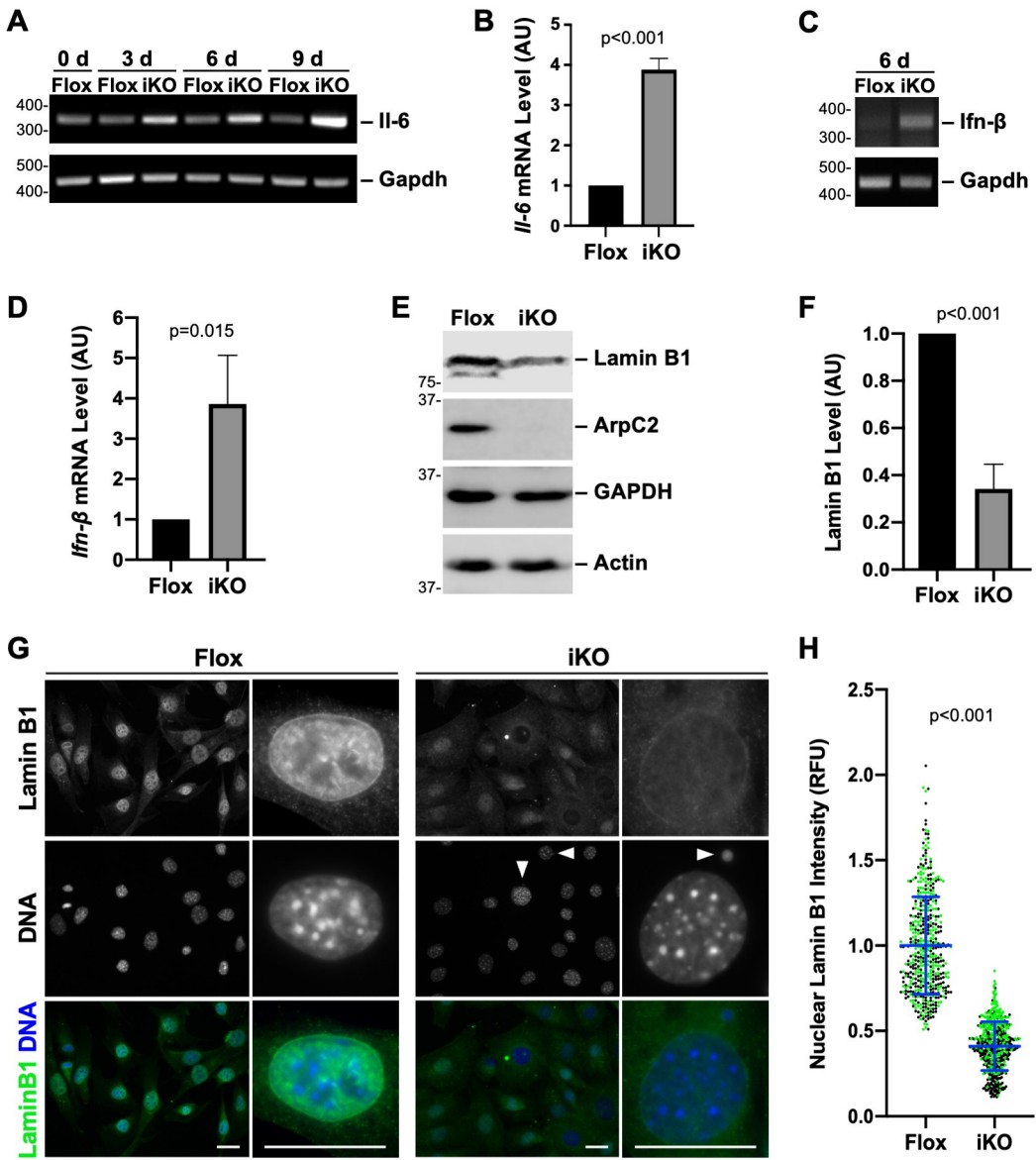

**Fig 2. Loss of the Arp2/3 complex results in elevated *Il-6* and *Ifn-β* transcript levels and decreased nuclear Lamin B1 expression.** (**A**) Mouse fibroblasts were treated with DMSO (Flox) or 4-OHT (iKO) for 0-6d and collected at 0, 3, 6, and 9d. RNA was isolated and RT-PCR performed using primers for *Il-6* and *Gapdh*. PCR products were visualized on ethidium bromide-stained agarose gels. (**B**) RT-qPCR was performed using primers for *Il-6* and *Gapdh* at 9d. *Il-6* product levels were normalized to *Gapdh*. Each bar represents the mean transcript abundance ±SD from n = 3 experiments. AU = Arbitrary Units. (**C**) RT-PCR was performed on Flox and iKO samples using primers for *Ifn-β* and *Gapdh* at 6d. (**D**) Agarose gel band intensities for *Ifn-β* were normalized to *Gapdh* intensities. Each bar represents the mean *Ifn-β* intensity ±SD from n = 3 experiments. (**E**) Flox and iKO cells were collected at 10d and immunoblotted with antibodies to Lamin B1, ArpC2, GAPDH, and actin. (**F**) Lamin B1 band intensities were normalized to GAPDH and actin band intensities. Each bar represents the mean intensity ±SD from n = 3 experiments. (**G**) Flox and iKO cells were fixed at 9d and stained with a Lamin B1 antibody (green) and DAPI (DNA; blue). Arrowheads point to micronuclei. Magnifications illustrate strong nucleoplasmic Lamin B1 localization for Flox cells. Scale bars, 25μm. (**H**) Nuclear Lamin B1 levels were quantified by outlining the DAPI-stained nucleus of each cell in ImageJ and measuring the mean Lamin B1 pixel intensity. Each dot represents an individual cell and the blue line depicts the average Lamin B1 pixel intensity ±SD from analyses of n = 492–556 cells pooled from 2 experiments (denoted in black or green). RFU = Relative Fluorescence Units.

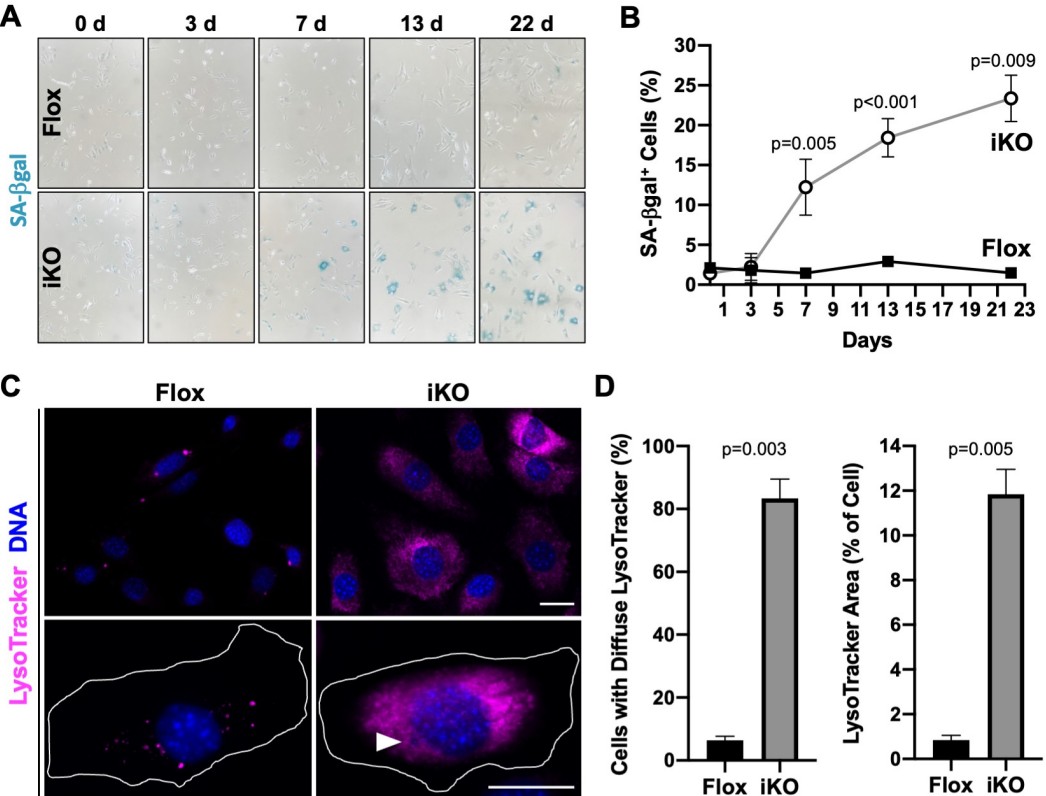

**Fig 3. SA-βgal activity and lysosomal staining are increased in ArpC2 iKO cells.** (A) Mouse fibroblasts were treated with DMSO (Flox) or 4-OHT (iKO) for 0-6d and senescence-associated beta-galactosidase (SA-βgal) staining was performed over a 22d period. (B) The % of SA-βgal-positive cells was quantified by calculating the number of intensely blue-colored cells divided by the total number of cells. Each point represents the mean % ±SD from a representative 0d timepoint, n = 2 experiments for the 3d and 22d timepoints, and n = 3 experiments for the 7d and 13d timepoints (727–2197 cells per point). (C) Flox and iKO cells at 9d were treated with LysoTracker (magenta), fixed, and stained with DAPI (DNA; blue). The magnifications illustrate punctate versus diffuse LysoTracker staining, and the arrowhead indicates the position of a micronucleus. Scale bars, 25μm. (D) The % of cells exhibiting diffuse LysoTracker staining was quantified by scoring the number of cells with broad instead of punctate cytoplasmic fluorescence and dividing by the total number of cells. The relative area within each cell occupied by LysoTracker was quantified using the threshold function in ImageJ. Each bar represents the mean % ±SD from n = 2 experiments (151–155 cells per bar).

to <1% of the area in Flox cells (Fig 3D). Thus, the cell cycle, morphological, and cytokine observations, when taken together with these Lamin B1, SA-βgal, and LysoTracker staining results, establish that ArpC2 iKO cells undergo senescence.

## The formation of micronuclei and DNA damage clusters precedes iKO cell senescence

Cellular senescence can be induced by a variety of stimuli, including DNA damage, oncogene activation, telomere shortening, and mitochondrial dysfunction [58–60]. Micronuclei and cytosolic chromatin fragments are common senescence-associated traits and influence the senescent state [53,61–63]. During the course of our characterization of the senescence parameters described above, we noticed that iKO cells frequently contained micronuclei whereas Flox cells did not (Figs 1C and 1H; 2G; 3C). Following specific examinations of samples containing DAPI-stained DNA, it became obvious that small cytoplasmic micronuclei were often

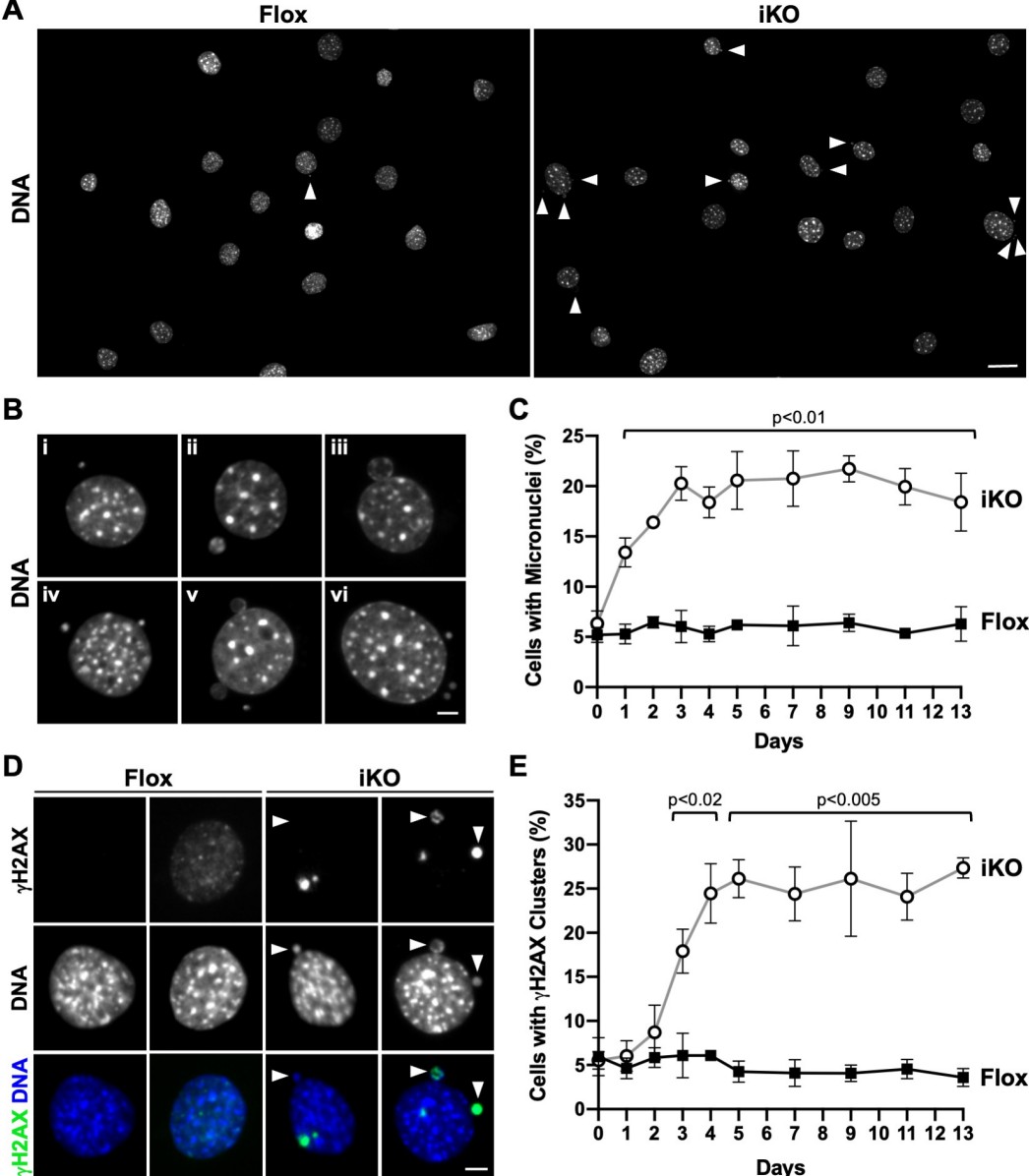

**Fig 4. Arp2/3 complex depletion leads to the biogenesis of cytoplasmic micronuclei and prominent clusters of DNA damage. (A)** Mouse fibroblasts were treated with DMSO (Flox) or 4-OHT (iKO) for 6d, fixed at 7d, and stained with DAPI (DNA). Arrowheads point to micronuclei. Scale bar, 25μm. **(B)** Magnifications show different micronucleus phenotypes in iKO cells. (vi) is the far-right cell from panel A. Scale bar, 5μm. **(C)** The % of cells with micronuclei was quantified over a 13d period following DMSO or 4-OHT exposure. Each point represents the mean % ±SD from n = 3 experiments, except for the 2d and 4d timepoints, which are from n = 2 experiments (432–631 cells per point in each experiment). **(D)** Flox and iKO cells were fixed at 7d and stained with a γH2AX antibody (green) and DAPI (blue). Magnifications show no or diffuse γH2AX staining in Flox nuclei and intense γH2AX clusters in iKO cell nuclei and micronuclei. Scale bar, 5μm. **(E)** The % of cells with γH2AX clusters was quantified over a 13d period following DMSO or 4-OHT exposure. Each point represents the mean % ±SD from n = 3 experiments, except for 2d and 4d timepoints, which are from n = 2 experiments (432–631 cells per point in each experiment).

present in iKO cells (Fig 4A). Of the iKO cells with micronuclei, most had one micronucleus (Fig 4B; i, ii, iii), but some harbored 2 or 3 micronuclei (Fig 4B; iv, v, vi). Micronuclei ranged in size and could be found completely detached (Fig 4B; i, ii, vi) or tethered to the periphery

(Fig 4B; iii, iv, v) of the main nucleus. Induction of CreER with 4-OHT in ArpC2-proficient mouse fibroblasts did not cause a proliferation arrest or micronucleus biogenesis (S3 Fig), indicating that these phenotypes were specific to cells that have lost the Arp2/3 complex.

We next quantified the timing and frequency with which micronuclei formed in Flox and iKO cells. Surprisingly, after just one day in 4-OHT, about 13% of iKO cells had micronuclei (Fig 4C). The percentage of iKO cells with micronuclei plateaued at approximately 20% by 3 days and remained steady out to 13 days, whereas the proportion of Flox cells with micronuclei stayed around 5% throughout the entire time course (Fig 4C). Therefore, a notably rapid formation of micronuclei precedes the proliferation arrest and SA-βgal positivity that results from *Arpc2* inactivation.

Micronuclei are often indicative of genomic instability, so we assessed the extent of DNA damage in Flox and iKO cells. Staining with an antibody to the phosphorylated histone protein H2AX (γH2AX), which is modified in response to double-stranded (ds) DNA breaks [64], demonstrated that iKO cells contained prominent DNA damage clusters in their nuclei and micronuclei (S4 Fig). Upon closer inspection, Flox cells typically exhibited no or diffuse γH2AX staining in their nuclei (Fig 4D), whereas iKO cells frequently had 1–3 very bright γH2AX clusters that localized to the nuclear periphery and/or within micronuclei (Fig 4D). Quantification of the percentage of cells with intense γH2AX clusters following the addition of DMSO or 4-OHT revealed that clusters began to increase in iKO cells by 2 days, and that from 3 days onward, clusters were significantly more common in iKO cells than in Flox cells (Fig 4E). The fraction of iKO cells with γH2AX clusters leveled out at approximately 25%, while the proportion of Flox cells containing clusters was always 3–6% (Fig 4E). These results extend previous observations in which *Drosophila* and mouse cells exposed to ionizing radiation and subjected to RNAi-mediated Arp2/3 depletion were found to contain DNA damage and micronuclei [27]. However, our results indicate that even without exposure to acute genotoxic agents, losing the Arp2/3 complex can cause an accumulation of damaged DNA elements that incorporate into micronuclei, thereby illustrating that the Arp2/3 complex is a crucial player in maintaining genomic integrity under relatively normal cell culture conditions.

## Arp2/3 complex inhibition causes DNA damage and proliferation arrest in other cells

To determine if inactivation of the Arp2/3 complex affected cell proliferation, micronucleus biogenesis, and DNA damage in other contexts, we treated multiple mouse cell lines with CK666, a pharmacological inhibitor of the complex [24,25]. Exposure of immortal NIH3T3 fibroblasts and B16-F1 melanoma cells to CK666 for 1.5 days resulted in a near-complete absence of mitotic cells (assessed by DNA and microtubule staining), a dramatic elevation in the population doubling times, a 2-3-fold increase in the proportion of cells with micronuclei, and a significant rise in nuclear γH2AX staining (S5 Fig). Similar results were also observed in human U2OS osteosarcoma cells, albeit using a higher concentration of CK666 and without the development of micronuclei (S6 Fig).

Since an increase in dsDNA breaks was a shared feature across all our manipulations of the Arp2/3 complex, we next tested whether introduction of ArpC2 into iKO cells could rescue their DNA damage phenotype. We expressed GFP or ArpC2-GFP in iKO cells, stained them for γH2AX, and found that the ArpC2-GFP-expressing cells had significantly less γH2AX staining than nearby non-transfected cells or GFP control cells (S7 Fig). Collectively, these observations demonstrate that the Arp2/3 complex is important for preventing the accumulation of dsDNA breaks in multiple cell types and that the specific ablation of ArpC2 causes the most severe outcomes in proliferation and DNA damage.

## Micronuclei form as a result of mitotic defects in ArpC2 iKO cells

Because ArpC2 iKO cells accumulated DNA damage and formed micronuclei with high frequency, we wanted to determine how the process of micronucleus biogenesis took place. Earlier studies have shown that some micronuclei form during mitosis as a result of chromosome missegregation, whereas other cytoplasmic chromatin fragments can arise during interphase following expulsion of DNA from the nucleus [61,65–68]. To better understand the mechanism of micronucleus formation in iKO cells and differentiate between the possibilities that micronuclei are a product of defects in mitosis versus nuclear remodeling in interphase, we expressed the GFP-tagged histone H2B in MTFs and visualized chromatin dynamics in live cells. These experiments were performed within the first 2 days of DMSO or 4-OHT treatment, when Arp2/3 complex levels were declining the fastest (Fig 1B), the cells were still dividing rapidly (Fig 1E), and the incidence of micronucleus formation was highest (Fig 4C).

Timelapse imaging of H2B-GFP-expressing Flox cells revealed that the majority of mitoses resulted in an equal partitioning of nuclear chromatin into two daughter cells (Fig 5A). However, iKO cell mitoses often yielded micronuclei due to errors in chromatin segregation (Fig 5A). This was repeatedly observed when chromatin fragments near the metaphase plate were not properly distributed to daughters during anaphase (Fig 5A). Another common phenotype of iKO cells that entered mitosis was premature mitotic exit (Fig 5B). Such cells underwent a prolonged prometaphase or metaphase, failed to enter anaphase, and ultimately returned to interphase with nuclei containing twice their normal chromatin content (Fig 5B). Micronuclei also formed in some of these cells that failed to complete mitosis (Fig 5B).

We next measured the frequencies with which the mitotic defects occurred. First, approximately 23% of completed mitoses in iKO cells yielded micronuclei compared to only about 6% of completed mitoses in Flox cells (Fig 5C). Second, after categorizing the stages of mitosis and determining the number of timelapse frames spent specifically in metaphase (judged by the presence of at least 95% of H2B-GFP fluorescence aligned compactly at the cell equator), we found that >80% of Flox cells that completed mitosis spent only one frame in metaphase, whereas only 40% of iKO cells proceeded through metaphase with this speed (Fig 5D). The other 60% of iKO cells that completed mitosis spent two or more frames in metaphase (Fig 5D), suggesting that iKO cells experience an unusually prolonged metaphase period. Third, when evaluating the incidence of premature mitotic exits (defined as cells that entered prophase but did not complete anaphase), we discovered that nearly 30% of iKO cells that entered prophase underwent premature mitotic exits compared to only 10% of Flox cells (Fig 5E). This observation indicates that the iKO population denoted "G2" in earlier FACS analyses (Fig 1F and 1G) was composed mostly of cells that possessed 4n DNA content because they failed to complete mitosis. Approximately 14% of premature mitotic exits that took place in the iKO cells also gave rise to micronuclei. Fourth, among the 330 iKO interphase nuclei that were observed during live cell imaging, at most 2 (i.e., ≤0.6%) appeared to create micronuclei via budding or blebbing, demonstrating that micronuclei form almost exclusively during mitosis.

As a final point, to confirm that the prevalent chromatin missegregation events in iKO cells were, at least in part, attributable to the presence of broken DNA fragments, we fixed and stained mitotic cells for γH2AX. In agreement with this possibility, during prometaphase and metaphase, iKO cells contained γH2AX foci at the periphery of or completely detached from the main chromatin mass (Fig 5F). Furthermore, in early anaphase, iKO cells displayed prominent DNA damage clusters near their equators, and by late anaphase, γH2AX-positive lagging chromosomes became apparent (Fig 5F). Overall, these live and fixed cell studies show that, in Arp2/3-depleted conditions, damaged DNA fragments persist and are incorporated into micronuclei as a result of defects in mitotic chromatin segregation.

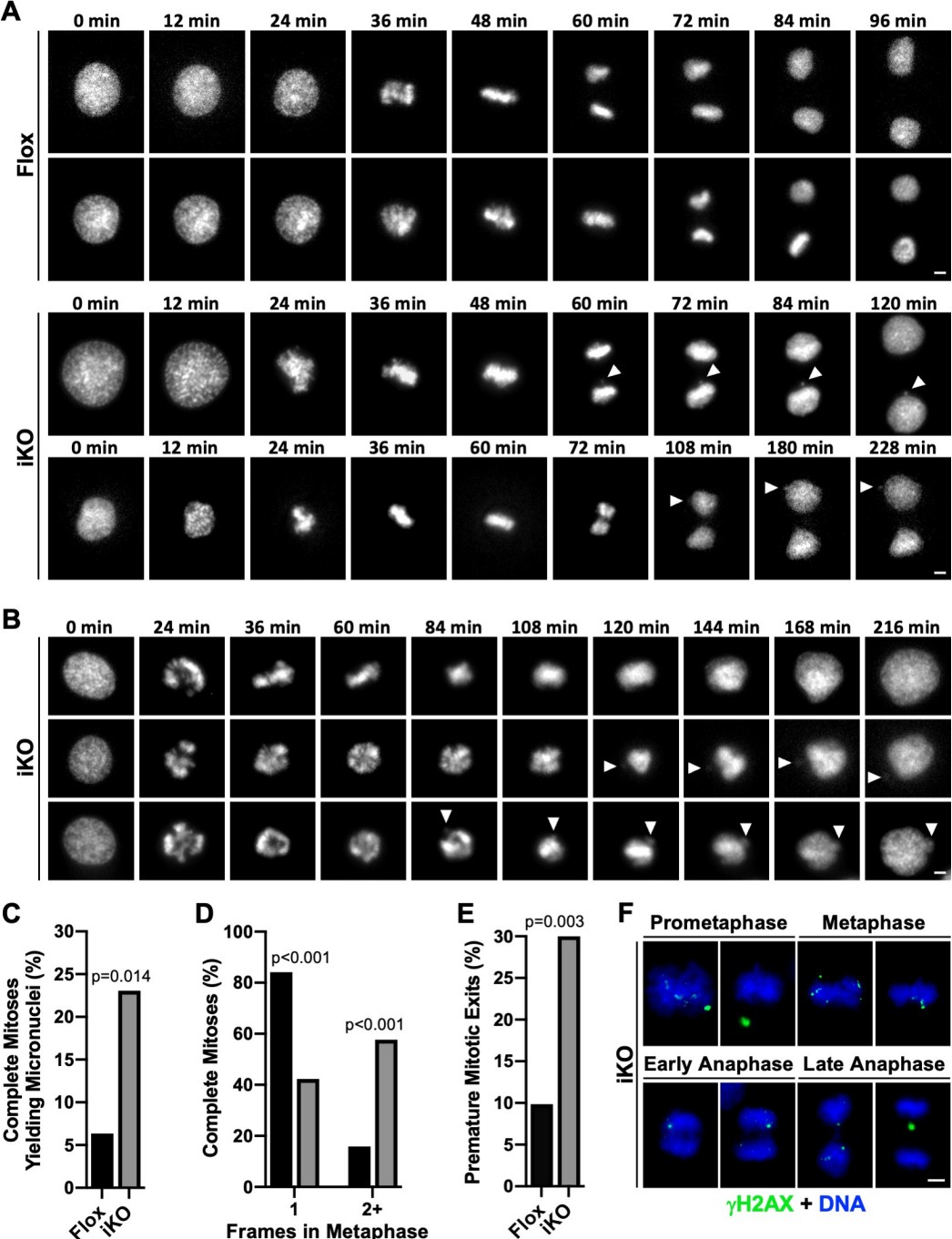

**Fig 5. Micronuclei form in ArpC2 iKO cells due to chromatin segregation errors and premature mitotic exits. (A-B)**
Live H2B-GFP-expressing mouse fibroblasts were imaged every 12min from 1-2d following DMSO (Flox) or 4-OHT (iKO)
exposure. Panel A shows completed mitoses, while panel B shows iKO cells exiting mitosis prematurely. Arrowheads in the
final 3–6 frames point to micronuclei biogenesis events. Scale bars, 5μm. **(C)** The % of completed mitoses yielding
micronuclei was calculated by dividing the number of mitoses yielding at least one micronucleus by the total number of
completed mitoses captured during live imaging (Flox n = 63; iKO n = 52; pooled from 13 Flox and 22 iKO experiments).
**(D)** All completed mitoses of Flox and iKO cells were binned into categories based on the number of timepoints observed in
metaphase. Black and grey bars represent Flox and iKO data, respectively. **(E)** The % of premature mitotic exits was
calculated by dividing the number of cells that entered prophase and returned to interphase without completing anaphase by
the total number of cells that entered prophase (Flox n = 71; iKO n = 70; pooled from 13 Flox and 22 iKO experiments). 10 of
the 70 iKO cells (14.3%) formed micronuclei. **(F)** iKO cells were fixed and stained with a γH2AX antibody (green) and DAPI
(DNA; blue). Movies for panels A and B appear in Supporting Information (S1–S7 Videos).

## Actin filament penetration into the central spindle is diminished in Arp2/3-depleted cells

Upon discovering that knocking out the Arp2/3 complex leads to chromosome partitioning defects, we wanted to also assess how the loss of this key actin nucleator might alter the actin and microtubule cytoskeletons during mitosis. Actin has been observed in various parts of meiotic and mitotic spindles in diverse organisms [30–39], so to examine the organization and intensity of actin filaments in relation to microtubule spindles in MTFs, we stained mitotic Flox and iKO cells with phalloidin to label F-actin, an anti-tubulin antibody to visualize microtubules, and DAPI to detect DNA. We focused on metaphase, when the chromosomes are either properly or improperly aligned at the central spindle, corresponding to when Flox mitoses rapidly proceed and iKO mitoses frequently stall.

Imaging of metaphase Flox cells in multiple focal planes revealed that several distinct F-actin structures were present. In the lower portions of cells, linear actin filaments in the shape of a spindle and several bundles of microtubules were observed (Fig 6A). In the middle planes of cells, multiple thick finger-like F-actin structures penetrated the chromosomal region, and numerous microtubules comprising the main microtubule spindle were apparent (Fig 6A). In the upper parts of cells, fewer F-actin and microtubule structures intercalated the central spindle area (Fig 6A). Staining with anti-ArpC2 antibodies indicated that the Arp2/3 complex was not enriched along the thin linear actin filaments at the lower part of the cell, but was present near F-actin and microtubules in the middle and upper spindle structures (Fig 6A). The presence of the Arp2/3 complex in the metaphase chromatin region was further verified using the ArpC2-GFP construct (S7 Fig) and antibodies to Arp3 (S8 Fig). Together, these observations expand the catalog of F-actin and Arp2/3-associated cytoskeletal structures that are found within dividing mammalian cells.

For more closely examining the spatial positioning of actin filaments, microtubules, and the Arp2/3 complex at the metaphase plate, we performed fluorescence intensity line scan analyses through the chromatin-containing region. Peaks of spindle microtubule intensity often coincided with peaks of F-actin intensity as well as sites of ArpC2 enrichment (Fig 6B). In contrast, DNA staining levels were highest where microtubules and F-actin were lowest (Fig 6B). These observations show a positive relationship between F-actin and microtubule localization amidst the chromosomes in the central metaphase spindle.

To determine whether loss of the Arp2/3 complex causes abnormalities in F-actin organization during metaphase, we compared F-actin staining in Flox and iKO cells. Actin filament levels in the metaphase chromatin-containing region appeared less prominent in iKO cells than in Flox cells (Fig 6C). Quantification of phalloidin fluorescence intensities in these regions of the central spindle revealed that F-actin levels were significantly lower in iKO cells compared to Flox cells (Fig 6D). This was not due to a general deficit in central spindle staining, because the fluorescence intensity of microtubules in the same region was not significantly different between iKO and Flox cells (Fig 6D). Moreover, metaphase F-actin abundance was not broadly diminished in iKO cells, as phalloidin staining around the centrosomal spindle poles did not significantly differ in iKO versus Flox cells either (S8 Fig). The specific reduction in the penetration of actin filaments, but not microtubules, into the metaphase chromatin-containing region of iKO cells was further reflected by fewer prominent F-actin peaks in fluorescence intensity plot profiles (Fig 6E). Collectively, these results indicate that a functional Arp2/3 complex is required for the polymerization of actin filaments in the vicinity of metaphase chromosomes.

Since metaphase spindle-associated F-actin overlapped with microtubules in Flox cells and was less prominent in Arp2/3 knockout cells, we considered that the subsequent arrangement

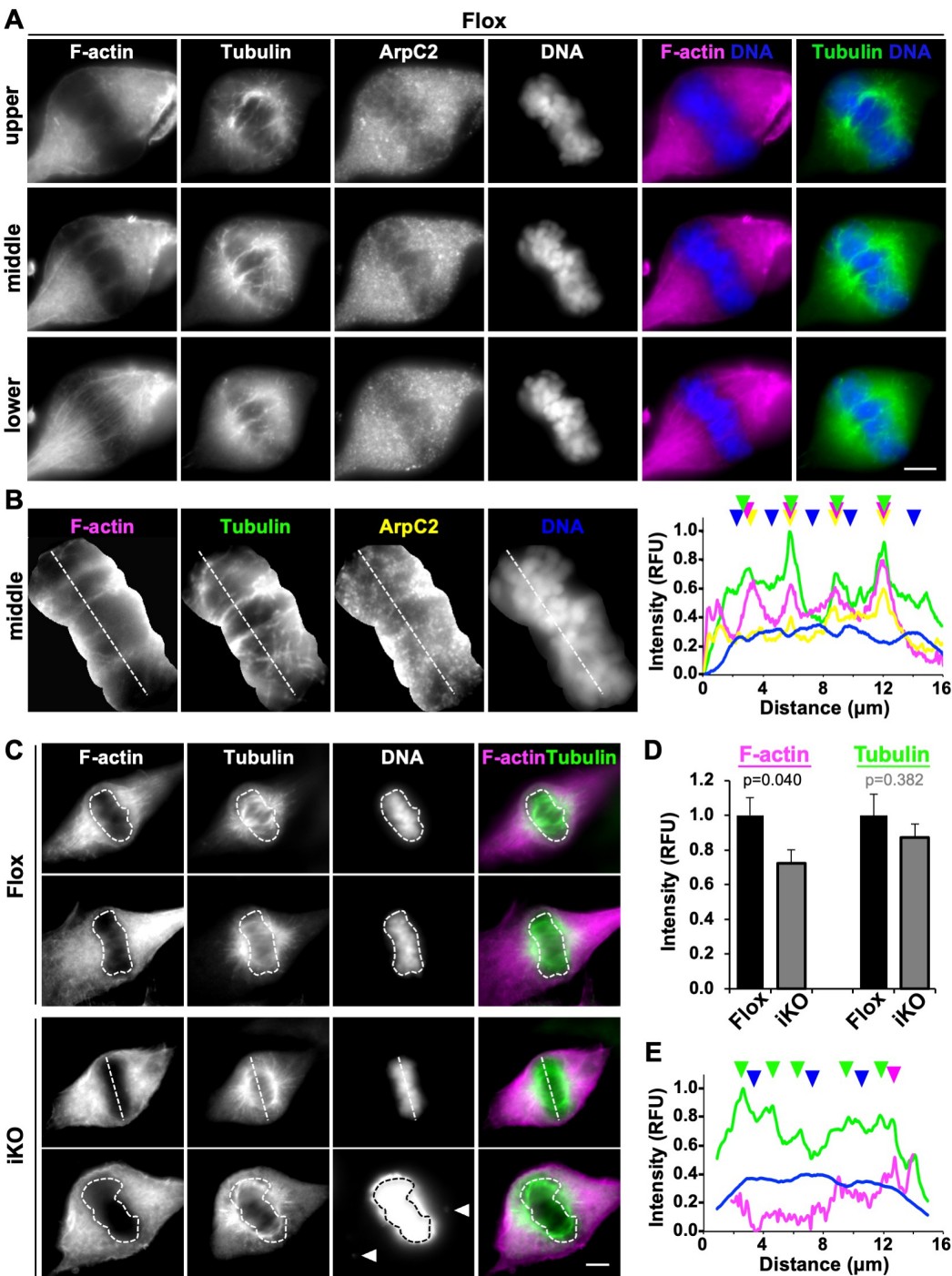

**Fig 6. Arp2/3 complex depletion reduces actin filament density in the chromatin-containing central spindle during metaphase. (A)** Mouse fibroblasts (Flox) were treated with DMSO for 1-2d, fixed, and stained with phalloidin (F-actin; magenta), an anti-tubulin antibody (green), anti-ArpC2 antibodies, and DAPI (DNA; blue). Images represent the upper, middle, and lower regions of metaphase cells. Scale bars, 5μm. **(B)** The DNA-containing region was isolated from the middle spindle in A, magnified, and subjected to line-scan fluorescence intensity analyses. Plot profiles depict the pixel intensity values for F-actin (magenta), tubulin (green), ArpC2 (yellow), and DNA (blue) along the dashed lines drawn across the representative metaphase region. Color-coded arrowheads highlight the overlapping peaks of F-actin, tubulin, and ArpC2 and the inverse intensity pattern for DNA. RFU = Relative Fluorescence Units. **(C)** Mouse fibroblasts were treated with DMSO (Flox) or 4-OHT (iKO) for 1-2d, fixed, and stained with phalloidin, an anti-tubulin antibody, and DAPI. The chromatin mass at the central metaphase spindle was outlined in ImageJ (dashed shapes) or a plot-profile line was drawn

through it (dashed line). The bottom DAPI-stained iKO cell panel was overexposed in order to draw attention to the presence of 2 chromatin fragments erroneously missing from the metaphase plate (arrowheads). **(D)** Fluorescence intensities of F-actin and tubulin were measured in chromatin areas outlined as in panel C. Each bar represents the mean intensity ±SD from n = 12 metaphase chromatin regions compiled from 3 experiments. **(E)** Plot profiles depict the pixel intensity values for tubulin, F-actin, and DNA along the dashed lines drawn across the metaphase region from the representative iKO cell in panel C.

of microtubules during chromosome segregation might be altered. To explore this possibility, we evaluated microtubule organization and intensity during anaphase, when the chromosomes are either correctly or incorrectly partitioned. Flox cells exhibited uniform distributions of microtubules across the width of the separating spindle (Fig 7A), as evidenced by the relatively evenly-spaced peaks in fluorescence intensity plots perpendicular to the presumed position of the cytokinetic ring (Fig 7B) and quantification of the left-right ratio of total tubulin staining in anaphase spindles (Fig 7C). In contrast, iKO cells displayed unbalanced tubulin intensities in which the microtubules appeared to be more heavily concentrated on one side of the spindle (Fig 7A–7C). Thus, while the presence of misplaced damaged chromatin fragments in iKO cells likely explains the biogenesis of micronuclei following mitosis, the accompanying decreases in actin filaments at the metaphase plate and alterations in anaphase microtubule organization are irregularities that may also influence chromosome missegregation and the formation of micronuclei in such Arp2/3-deficient cells.

## p53-associated upregulation of p21 is responsible for the cell cycle arrest in iKO cells

Considering the prevalence of γH2AX staining, broken chromatin fragments, and micronuclei in Arp2/3 complex-deficient cells, we hypothesized that the underlying mechanism for driving senescence in these cells is a DNA damage response. To explore this possibility, we immunoblotted Flox and iKO cell extracts with antibodies to the crucial tumor suppressor protein and transcription factor p53, which typically becomes stabilized, phosphorylated, and enriched in the nucleus following DNA damage [69,70]. Consistent with this expectation, by 3 days after 4-OHT vs DMSO treatment, p53 levels were higher in iKO cells than in Flox cells, and p53 was phosphorylated on serine15 (Fig 8A). Quantification indicated that, in iKO cells, total p53 levels had doubled by 6 days and tripled by 9 days (Fig 8B). Furthermore, immunofluorescence microscopy showed that nuclear p53 fluorescence was more intense in iKO cells than in Flox cells (Fig 8C). Phosphorylated p53 was also enriched in the nuclei of iKO cells (Fig 8D), where it became >5-fold more abundant than in Flox cells (Fig 8E). These results imply that a p53-mediated DNA damage response is induced in Arp2/3-depleted cells.

Two of the major factors involved in the cell cycle arrest that leads to senescence are the cyclin-dependent kinase inhibitors *Cdkn2a*/p16INK4A (p16) and *Cdkn1a*/p21CIP/WAF (p21). Elevated levels of both of these anti-proliferative transcripts/proteins are frequently used as indicators of the senescent state [58,71], although populations of cells expressing high levels of p16 appear to be distinct from those expressing high levels of p21, at least in tissues from aged mice [72]. Since p21 is a well-known transcriptional target of p53 [73,74], and the MTFs used in our studies lack p16, we postulated that *Cdkn1a*/p21 was associated with the onset of senescence in the iKO cells. To test this, we performed RT-PCR (Fig 9A) and RT-qPCR (Fig 9B) to compare *Cdkn1a* transcript levels in Flox and iKO cells over a 9-day period. Indeed, *Cdkn1a* expression appeared greater in the iKO cells than in the Flox cells at 6 and 9 days after the initiation of 4-OHT treatment (Fig 9B). RT-qPCR revealed that *Cdkn1a* transcript levels were doubled in the iKO cells relative to the Flox cells at 9 days (Fig 9B), showing that deletion of the Arp2/3 complex leads to an upregulation of this key cell cycle regulator.

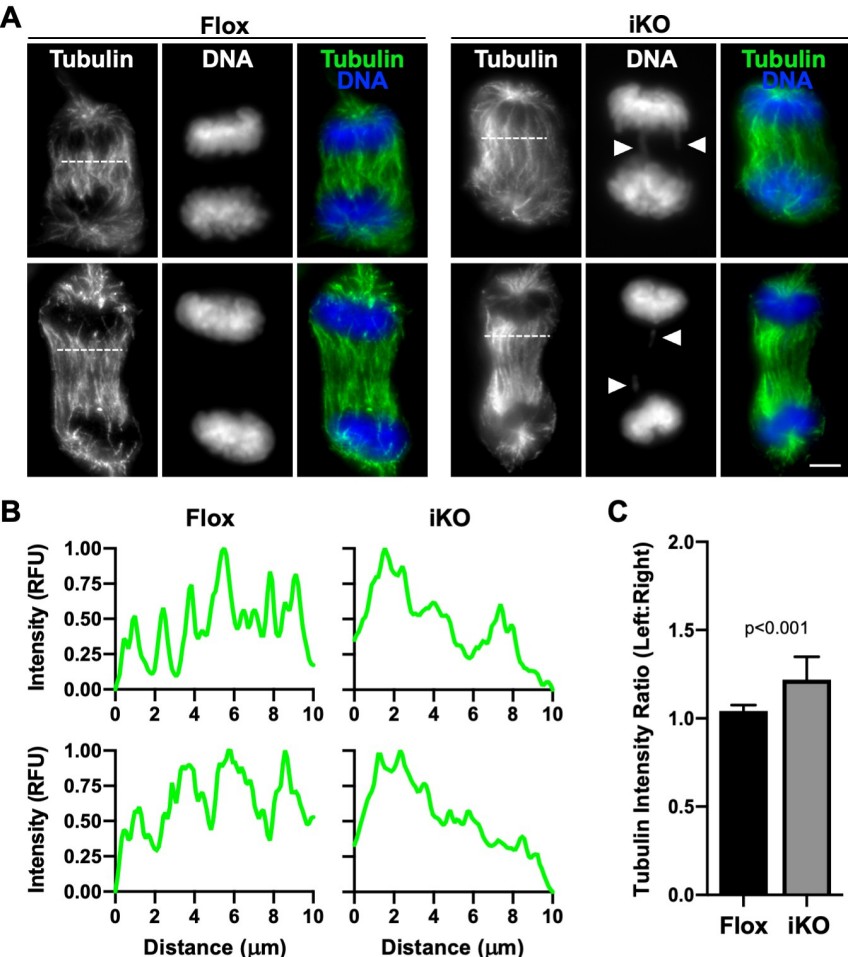

**Fig 7. Anaphase microtubule organization is unbalanced in ArpC2 iKO cells. (A)** Mouse fibroblasts were treated with DMSO (Flox) or 4-OHT (iKO) for 1d, fixed, and stained with a tubulin antibody (green) and DAPI (DNA; blue). Arrowheads highlight sites of aberrant chromatin localization. Scale bar, 5μm. **(B)** Line-scan plot profiles depict the pixel intensity of tubulin along the horizontal dashed lines drawn across the representative mitotic spindles from panel A. RFU = Relative Fluorescence Units. **(C)** Lines drawn perpendicular to the dashed lines were used to divide anaphase central spindles into left and right halves for measuring fluorescence intensities of microtubules. Each bar represents the mean left:right tubulin intensity ratio ±SD of n = 16 anaphase cells.

To evaluate whether the increase in *Cdkn1a* transcript corresponded to greater p21 protein levels, Flox and iKO cells were treated with antibodies to p21 and subjected to immunofluorescence microscopy (Fig 9C). Quantification of p21 nuclear intensity at the 6 day timepoint verified that p21 levels were significantly higher in iKO than in Flox cells (Fig 9C and 9D). Together, the upregulation and nuclear localization of p53 and p21 in iKO cells support the idea that a cell cycle arrest pathway is activated after the Arp2/3 complex is removed.

To assess the functional importance of *Cdkn1a*/p21 in the proliferation arrest that takes place upon Arp2/3 depletion, we added DMSO or 4-OHT to MTFs and transfected the cells with siRNAs targeting *Cdkn1a* on day 3, just before the iKO cells would normally begin to arrest. We then quantified p21 protein knockdown, the proportion of cells in mitosis, and the cell population doubling times after day 6. As expected (Fig 9C and 9D), in experiments using control siRNAs, more p21 was present in iKO cells than in Flox cells (Fig 9E and 9F). Under

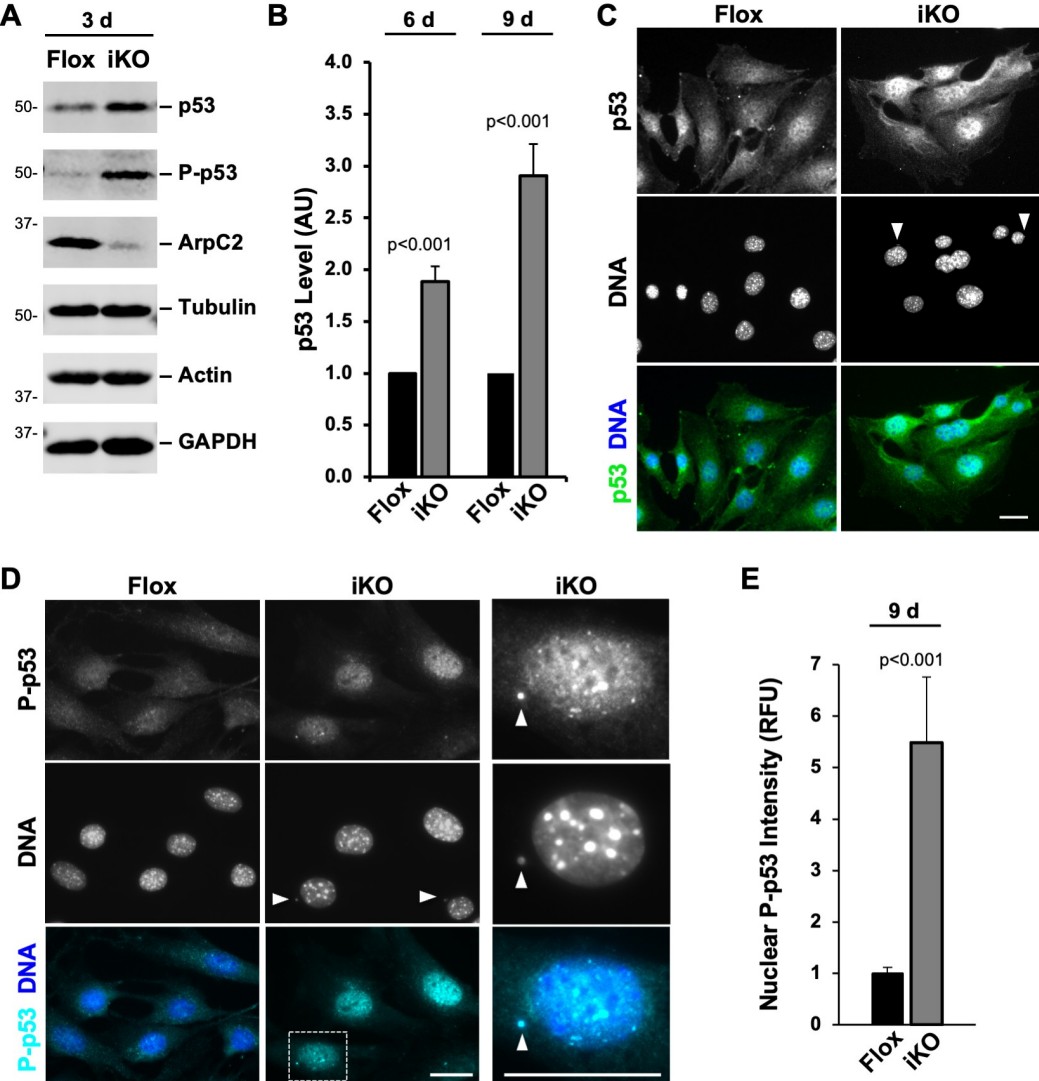

**Fig 8. A p53 response is induced when the Arp2/3 complex is deleted.** (A) Mouse fibroblasts were treated with DMSO (Flox) or 4-OHT (iKO) for 3d, collected, and immunoblotted with antibodies to total p53, Phospho-serine15 of p53 (P-p53), ArpC2, tubulin, actin, and GAPDH. (B) Flox and iKO cells were collected at 6 or 9d and immunoblotted. p53 band intensities were normalized to tubulin, actin, and GAPDH band intensities. Each bar represents the mean intensity ±SD from n = 3 or 4 experiments. (C) Flox and iKO cells were fixed at 3d and stained with a p53 antibody (green) and DAPI (DNA; blue). Arrowheads point to micronuclei. Scale bars, 25μm. (D) Flox and iKO cells were fixed at 9d and stained with an antibody to P-p53 (cyan) and DAPI. (E) Nuclear P-p53 levels were quantified by outlining the DAPI-stained nucleus of each cell in ImageJ and measuring the mean P-p53 pixel intensity. Each bar represents the mean % ±SD from n = 22–32 cells per bar pooled from multiple experiments. RFU = Relative Fluorescence Units.

these conditions, around 4% of Flox cells were observed to be in mitosis compared to 0% of iKO cells (Fig 9G). Moreover, similar to earlier experiments (Fig 1E), the population doubling time was <20h in Flox cells and >70h in iKO cells (Fig 9H). Parallel treatments of cells with two independent siRNAs to *Cdkn1a* caused a modest reduction of the already-low p21 protein levels in Flox cells and prevented the upregulation of p21 in iKO cells (Fig 9E and 9F). Phenotypically, the targeting of *Cdkn1a* did not affect the number of mitotic Flox cells (Fig 9G) or the Flox population doubling time (Fig 9H). In contrast, RNAi of *Cdkn1a* in iKO cells

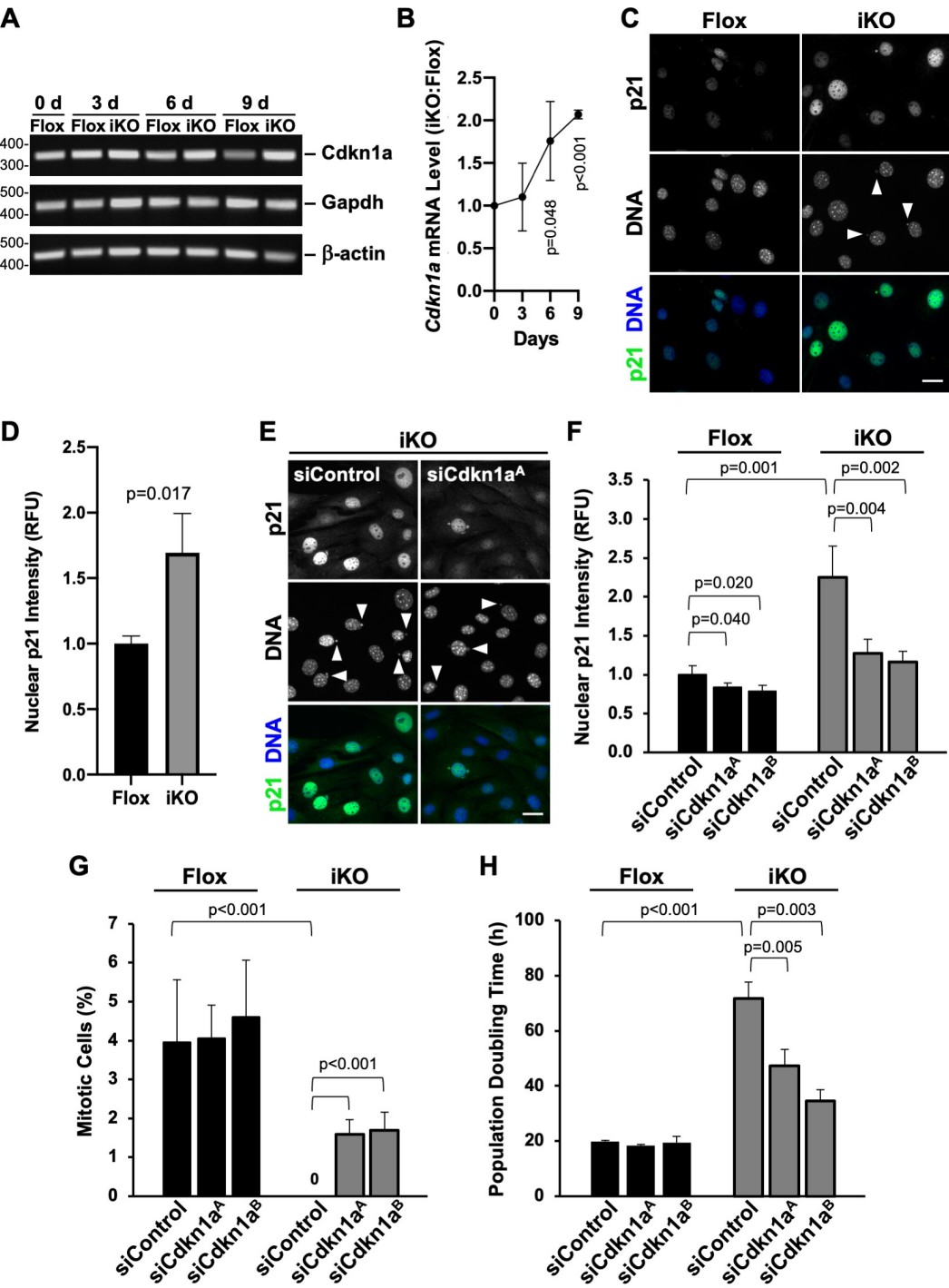

**Fig 9. Activation of *Cdkn1a*/p21 is responsible for the cell cycle arrest in ArpC2 iKO cells. (A)** Mouse fibroblasts were treated with DMSO (Flox) or 4-OHT (iKO) for 0-6d and collected at 0, 3, 6, and 9d. RNA was isolated and RT-PCR performed using primers for *Cdkn1a*, *Gapdh*, and *β-actin*. PCR products were visualized on ethidium bromide-stained agarose gels. **(B)** RT-qPCR was performed using primers for *Cdkn1a* and *Gapdh* at 0, 3, 6, and 9d. *Cdkn1a* product levels were normalized to *Gapdh* and plotted as the iKO:Flox ratio. Each point represents the mean ratio ±SD from n = 3 experiments. **(C)** Flox and iKO cells were fixed at 6d and stained with a p21 antibody (green) and DAPI (DNA; blue). Scale bars, 25μm. **(D)** Nuclear p21 fluorescence was quantified by outlining the DAPI-stained nucleus of each cell in ImageJ and measuring the mean p21 pixel intensity. Each bar represents the mean intensity ±SD from n = 3 experiments (653–658 cells per bar). RFU = Relative Fluorescence Units. **(E)** Flox and iKO cells were treated with control siRNAs or independent siRNAs for the *Cdkn1a* gene on day 3 and fixed on day 6 before performing immunofluorescence as in C. **(F)** Nuclear p21

fluorescence was quantified as in D from n = 4 experiments (898–1820 cells per bar). **(G)** The % of cells in mitosis was quantified for samples in E-F. Each bar represents the mean % ±SD from n = 4 experiments. **(H)** Flox and iKO cells were treated with control siRNAs or independent siRNAs for the *Cdkn1a* gene on day 3 and population doubling times were quantified from 5-7d. Each bar represents the mean doubling time ±SD from n = 3 experiments.

significantly increased the number of mitotic cells (Fig 9G) and reduced the population doubling time (Fig 9H). Together, these results demonstrate that transiently blocking the upregulation of p21 enables nearly half of the iKO cells to continue replicating. *Cdkn1a*/p21 therefore appears to be a primary player during the induction of the cell cycle arrest in Arp2/3 complex knockout cells.

## Signaling via cGAS, STING, and IRF3 affects a subset of ArpC2 iKO cells

In addition to the above nuclear changes that took place upon Arp2/3 ablation, it seemed likely that cytoplasmic changes arising from the presence of micronuclei in ArpC2 iKO cells could also be linked to the senescent state. We hypothesized that a cytosolic DNA detection and signaling pathway involving the cyclic GMP-AMP Synthase (cGAS) enzyme, which recognizes extra-nuclear chromatin and relays a signal to the downstream effector molecule STING [53,63, 75,76], might also be activated in iKO cells. Tagged cGAS can be recruited to micronuclei, and through its detection of cytosolic DNA and activation of STING, promotes pro-senescence and pro-inflammatory gene expression, including an interferon response [77–79]. To determine if tagged cGAS localizes to the micronuclei in ArpC2 iKO cells, we transiently transfected MTFs with plasmids encoding mCherry-cGAS or mCherry as a control (S9 Fig) and treated them with 4-OHT to induce the deletion of *Arpc2*. mCherry was highly expressed in the iKO cells but was not recruited to micronuclei (Fig 10A). In contrast, mCherry-cGAS showed intense localization to micronuclei (Fig 10A), indicating that it can detect the cytosolic DNA in iKO cells.

We next wanted to determine the localization of endogenous cGAS and STING in the MTFs. Immunoblotting indicated that cGAS and STING were expressed at similar levels in Flox and iKO cells (Fig 10B and 10C), so to test whether cGAS and/or STING were recruited to the micronuclei, we visualized these proteins via immunofluorescence microscopy after 3–4 days, when micronuclei become more abundant in iKO cells and just before Flox and iKO multiplication rates begin to diverge. Consistent with previous studies of other cells [80–82], endogenous cGAS was present in the nuclei of both Flox and iKO cells (Fig 10D). It was also associated with nearly 40% of iKO cell micronuclei (Fig 10D). The rare micronuclei that formed in Flox cells also recruited cGAS (Fig 10D magnification).

For STING, a membrane protein that localizes to organelles of the conventional secretory pathway including the endoplasmic reticulum (ER) and Golgi [82], antibody staining revealed a speckled ER-like localization in both Flox and iKO cells, and enrichment around some micronuclei (Fig 10D inset). Immunolabeling for the ER chaperone GRP94 and the *cis*-Golgi protein GM130 implied that ER but not Golgi membranes are more likely to surround micronuclei in iKO cells (Fig 10E). These findings are consistent with a function for cGAS in recognizing damaged DNA in the cytosol and initiating local recruitment of ER-associated STING near micronuclei in Arp2/3-deficient cells.

Upon activation by cGAS-mediated cGAMP synthesis, STING re-localizes from the ER to the Golgi, where it is phosphorylated by the protein kinase TBK1, leading to the phosphorylation and activation of the transcription factor IRF3, which induces a type-I interferon response [83–87]. Phospho-serine365 of STING (P-STING) is crucial for such interferon induction pathways in mice [88]. Our earlier observations that *Ifn-β* is upregulated and that cGAS and

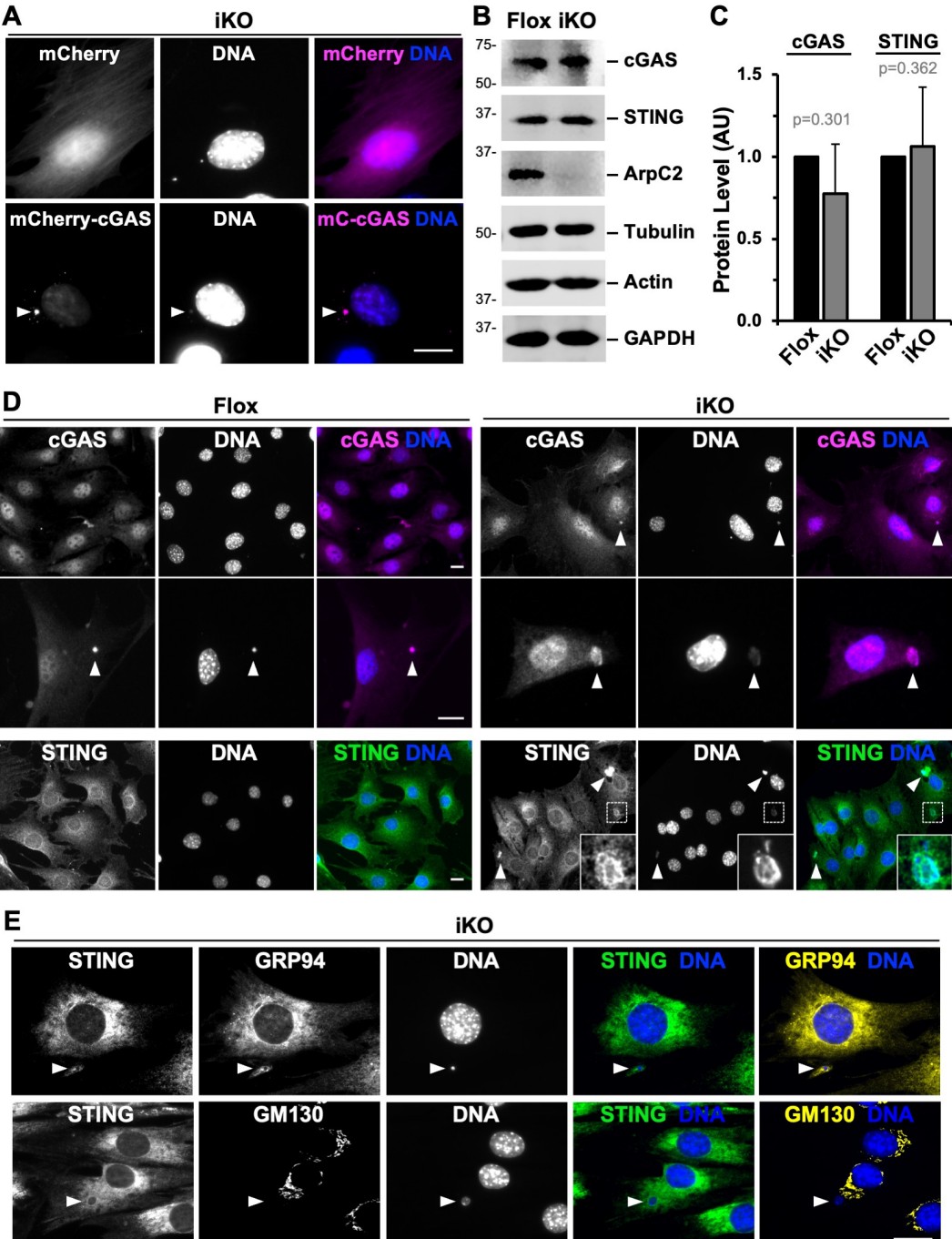

**Fig 10. cGAS and STING are recruited to micronuclei. (A)** Mouse fibroblasts were transfected with plasmids encoding mCherry or mCherry-cGAS (magenta), treated with 4-OHT (iKO), fixed at 1-2d, and stained with DAPI (DNA; blue). Arrowheads point to a cGAS-positive micronucleus. Scale bars, 20μm. **(B)** Cells were treated with DMSO (Flox) or 4-OHT (iKO) for 3-4d and immunoblotted with antibodies to cGAS, STING, ArpC2, tubulin, actin, and GAPDH. **(C)** cGAS and STING band intensities were normalized to tubulin, actin, and GAPDH band intensities. Each bar represents the mean intensity ±SD from n = 3 or 6 experiments. **(D)** Flox and iKO cells were fixed at 3d and stained with cGAS (magenta) or STING (green) antibodies and DAPI. Arrowheads point to cGAS- or STING-positive micronuclei. 38.7% of micronuclei in iKO cells were cGAS-positive (75 cells evaluated from 2 experiments). Insets show STING localization around an iKO micronucleus. **(E)** iKO cells were fixed at 6d and stained with antibodies to STING (green) and either GRP94 or GM130 (yellow).

STING can be recruited to micronuclei upon Arp2/3 complex depletion led us to explore whether cGAS affects STING and IRF3 activation when iKO cells initiate senescence. MTFs were exposed to 4-OHT for 3 days to induce the deletion of *Arpc2* and then transferred into media containing the cGAS inhibitor RU.521 [89] or DMSO as a control. In Flox cells, P-STING staining was weak and diffuse, irrespective of DMSO or RU.521 exposure (Fig 11A), suggesting that STING is generally inactive in those cells. On the contrary, in DMSO-treated iKO cells harboring micronuclei, P-STING was detectable at the Golgi, a phenotype that could be prevented by the 3-day administration of RU.521 (Fig 11A).

IRF3 is imported from the cytosol into the nucleus to perform its transcription factor functions, so as a readout of signaling to IRF3 we measured its nuclear intensity by immunofluorescence. In Flox cells, IRF3 staining was both cytosolic and nuclear whether or not RU.521 was added (Fig 11B). In contrast, in DMSO-treated iKO cells, the nuclear intensity of IRF3 was visibly increased (Fig 11B). Quantification revealed that nuclear IRF3 staining averaged 30% higher in iKO cells than in Flox cells (Fig 11C), and that the ratio of nuclear-to-cytoplasmic IRF3 intensity was significantly greater in iKO cells (Fig 11D). The scoring of cells possessing normal amounts of nuclear IRF3 versus those with noticeably high amounts of nuclear IRF3 (IRF3$^{HIGH}$ cells) indicated that >3-fold more iKO than Flox cells were classified as IRF3$^{HIGH}$ (Fig 11C). Similarly, the scoring of cells with significantly higher nuclear than cytoplasmic IRF3 intensity (IRF3$^{N>C}$ cells) revealed that >20% of iKO cells but 0% of Flox cells were categorized as IRF3 nuclear-enriched. Exposure of iKO cells to RU.521 suppressed the IRF3$^{HIGH}$ phenotype and abolished the IRF3$^{N>C}$ phenotype (Fig 11C and 11D), showing that inhibition of cGAS was able to diminish signaling to STING and IRF3 in iKO cells.

Finally, to evaluate whether cGAS signaling influences the cell proliferation arrest that takes place when the Arp2/3 complex is lost, cell replication rates were measured in the absence or presence of RU.521 during days 4 and 5 following the onset of 4-OHT treatment. RU.521 caused a modest but statistically faster population doubling time in the iKO cells (Fig 11E), implying that cGAS inhibition has the capacity to oppose the initiation of senescence in some of the cells in this experimental system. Collectively, our cGAS localization, STING phosphorylation, IRF3 accumulation, and pharmacological inhibitor results support the idea that cGAS-STING-IRF3 signaling is a secondary contributor to the establishment of senescence in Arp2/3-deficient cells.

## Discussion

The Arp2/3 complex is a key driver of many cellular processes that require actin assembly at the plasma membrane, namely adhesion, endocytosis, protrusion, and migration. While the complex is evolutionarily conserved among nearly all eukaryotes and essential for viability in animals, the molecular basis underlying its indispensability is unclear. Roles for the Arp2/3 complex in enabling DNA repair during interphase, promoting chromosome partitioning in meiosis/mitosis, and controlling apoptosis following DNA damage have recently emerged and may help explain why Arp2/3 is essential. Our current results provide additional insights into how the mammalian Arp2/3 complex maintains genomic integrity and supports mitotic progression. We show that deletion of the Arp2/3 complex has multiple significant molecular consequences–beginning with unrepaired DNA damage and including spindle actin and microtubule abnormalities–that lead to the biogenesis of micronuclei, p53 activation, p21-mediated cell cycle arrest, cGAS/STING signaling, and cellular senescence (S10 Fig).

Various endogenous stresses are known to induce cellular senescence, including replicative, telomeric, genotoxic, oncogenic, oxidative, and mitochondrial stress [90]. Our studies now

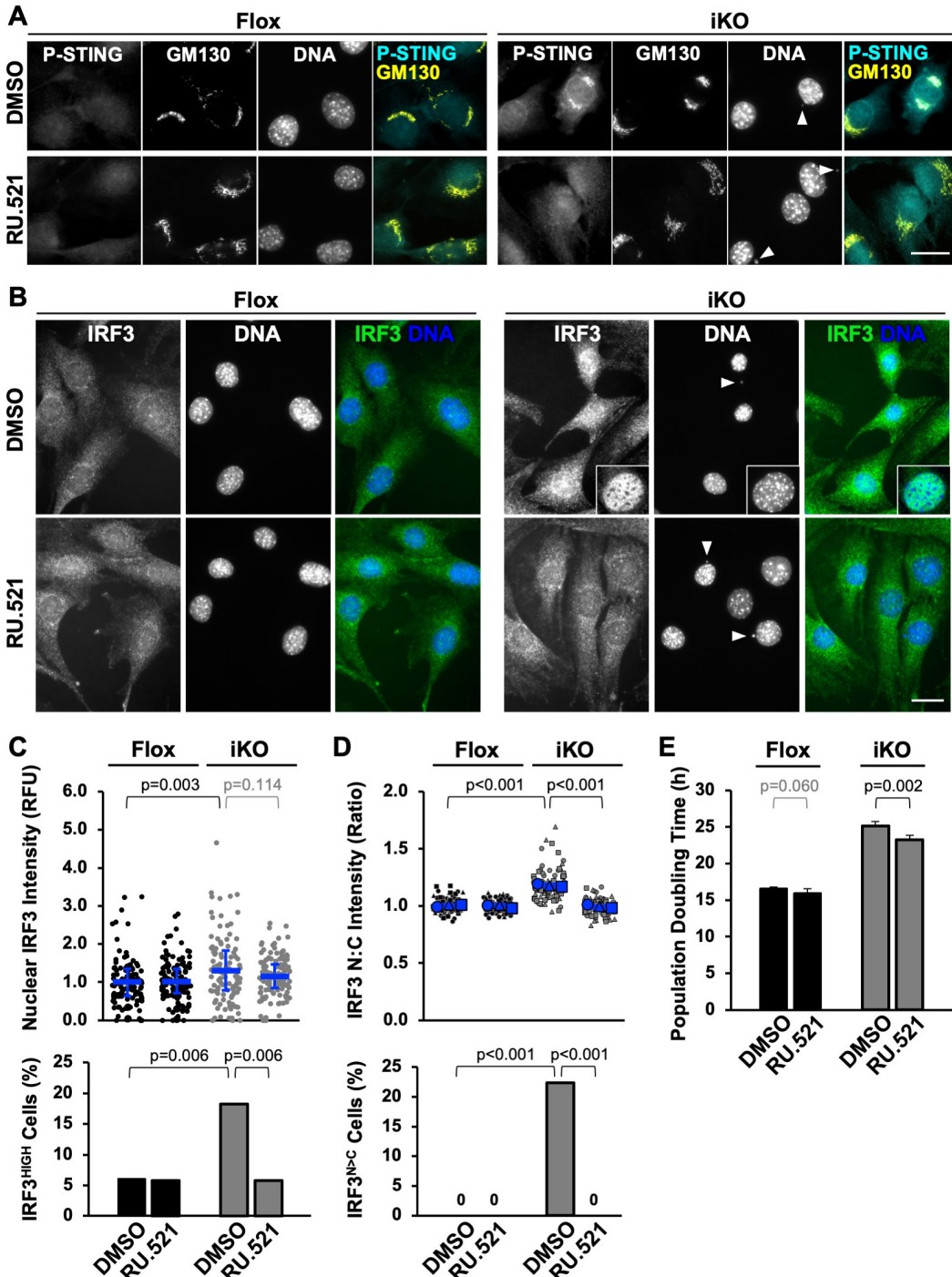

**Fig 11. cGAS-associated activation of STING-IRF3 signaling affects the proliferation arrest of ArpC2-deficient cells. (A)** Mouse fibroblasts were treated with DMSO (Flox) or 4-OHT (iKO) for 3d, switched to media containing DMSO or RU.521, fixed at 6d, and stained with antibodies to Phospho-serine365 of STING (P-STING; cyan) and GM130 (yellow), and with DAPI (DNA; blue). Arrowheads point to micronuclei. Scale bars, 25μm. **(B)** Cells were treated as in A and stained with IRF3 antibodies (green) and DAPI. The intranuclear staining pattern of IRF3 is highlighted in a magnified inset. **(C)** Nuclear IRF3 fluorescence was quantified by outlining the DAPI-stained nucleus of each cell in ImageJ and measuring the mean IRF3 pixel intensity. Each dot represents an individual cell and the blue lines represent the mean intensities from analyses of n = 104–120 cells pooled from 3 experiments. RFU = Relative Fluorescence Units. IRF3 highly-expressing (IRF3$^{HIGH}$) cells were defined as those with nuclear expression levels >2-fold higher than the mean nuclear IRF3 fluorescence of Flox cells. **(D)** The ratio of nuclear (N) to cytoplasmic (C) IRF3 fluorescence intensity was quantified. Each dot represents an individual cell, the

different symbols indicate different experiments, the large blue symbols depict the means from each experiment, and the blue lines show the mean ratio ±SD from n = 3 experiments (100–103 cells per condition). IRF3$^{N>C}$ cells were defined as those with IRF3 nuclear-to-cytoplasmic ratios >1.25. (**E**) Cells were treated with DMSO or 4-OHT for 3d, switched to media containing DMSO or RU.521, and population doubling times were quantified between 4d and 6d. Each bar represents the mean doubling time ±SD from n = 5 experiments.

add cytoskeletal dysfunction as a trigger of senescence. Arp2/3 complex knockout cells, in addition to a stable proliferation arrest and physical enlargement, display multiple biomarkers of senescence. These include an increase in *Il-6* and *Ifn-β* transcripts and a reduction of nuclear Lamin B1 levels. The iKO cells also harbor high cytoplasmic SA-βgal activity and increased acidic organelle content. Given that the Arp2/3 complex influences most cellular functions, from plasma membrane remodeling [2,15,91] to mitochondrial dynamics [92] to autophagy [93] to DNA repair [27,28], it seems likely that multiple intracellular defects derived from Arp2/3 deficiency can impact senescence induction. Potential changes in mitochondrial turnover combined with altered lysosomal function could signify increases in mitochondrial, oxidative, and proteotoxic stress in iKO cells. The extent to which these stressors contribute to the initiation or maintenance of senescence will be an important area for future investigation.

Despite uncertainties in the severity of cytoplasmic organelle dysfunction in iKO cells, obvious defects in nuclear and chromatin-associated processes emerged as major factors in promoting senescence in our studies (S10 Fig). Kinetic analyses indicate that 1–3 days after exposure to 4-OHT, ArpC2 (and thus functional Arp2/3 complex) levels are 50–90% depleted (Fig 1). During this time window, such cells are actively proliferating, but are plagued by mitotic errors due to the presence of damaged DNA and irregularities in actin and microtubule spindle organization (Figs 5, 6, and 7). This period therefore coincides with the steepest increases in detection of micronuclei via DAPI staining and visualization of dsDNA breaks in both nuclei and micronuclei via γH2AX staining (Fig 4). A DNA damage response is then activated beginning in the 3–6 day range, as evidenced by increased p53 expression and phosphorylation (Fig 8). By 6 days, mitotic cells are absent, as the p53-regulated cell cycle inhibitor p21 accumulates in the nucleus where it plays an important role in inhibiting cell multiplication (Fig 9). At this stage, micronuclei elicit an interferon response via cGAS/STING signaling, which further opposes proliferation (Figs 2, 10, and 11). After 7–10 days, the entire population of iKO cells appears to be senescent, given the complete lack of multiplication (Fig 1), as well as single cell phenotypes in enlargement (Fig 1), nuclear Lamin B1 reduction (Fig 2), and SA-βgal-staining concomitant with cytoplasmic acidity (Fig 3). From 10 days onward, the iKO cell population remains viable but non-replicative while maintaining 20–25% positivity for micronuclei and SA-βgal.

While all Arp2/3 knockout cells senesce, it is unclear why only a fraction of the culture is SA-βgal positive. This could be a technical matter of assay sensitivity, be related to lysosomal dysfunction, or SA-βgal activity may simply develop later in some cells. Additionally, the reasons why the entire culture senesces when only a quarter of the iKO cells harbor detectable micronuclei remain somewhat ambiguous. Many iKO cells without micronuclei are P-p53 or p21 positive, so it is conceivable that p53-mediated arrest programs are activated in micronucleus-negative cells due to aneuploidy (e.g., insufficient chromatin following defective mitoses or excess DNA content following premature mitotic exits). Paracrine signaling from mature senescent cells to others in the culture may also support the global proliferation block.

In response to damaging stimuli, the cyclin-dependent kinase inhibitors p16 and p21 are frequently upregulated and act as key players in pathways that promote the cell cycle arrest that defines senescence [58–60]. Early attempts at establishing Arp2/3-depleted mouse fibroblast cultures indicated that inactivation of *p16Ink4a/Arf* was necessary for cell proliferation,

suggesting that the loss of Arp2/3 can induce a growth arrest or cell death in a p16-mediated manner [40,41]. Subsequent experiments demonstrated that transient Arp2/3 complex inhibition using CK666 prevents DNA replication in multiple mammalian cell lines and can cause MCF10A epithelial cells to undergo a temporary p21-dependent G1 arrest [94]. Our current work shows that CK666 can increase DNA damage and block the proliferation of several other mammalian cell lines as well.

Additional studies in human cells treated with a genotoxic agent indicate that depletion of the Arp2/3 complex or inactivation of two of its upstream regulators impairs the execution of apoptosis [29]. Such cells undergo a p53- and p21-associated cell cycle arrest but fail to properly complete an apoptosome-based caspase cleavage cascade [29]. All of the above observations, together with our data showing that ArpC2 iKO cells activate p53 and p21 in a p16-independent manner, suggest that losing the Arp2/3 complex can trigger multiple cell cycle arrest pathways. While p53 is known to cause upregulation of *Cdkn1a*/p21 to promote arrest [95–97], deciphering the degree to which other anti-proliferative mechanisms cooperate with p21 in iKO cells requires more investigation. Interestingly, studies in aged mice show that cells expressing high levels of p21 are mostly distinct from those expressing high levels of p16 [72], further highlighting the possibility that loss of the Arp2/3 complex *in vivo* may instigate different pathways to senescence depending on the physiological state of the cell and its tissue of origin.

In addition to the functions of p53 and p21 in the nucleus, the recognition of micronuclei in the cytoplasm affects the senescence-related phenotypes in Arp2/3-deficient cells. We observed cGAS and STING localization at or around micronuclei in ArpC2 iKO cells, nuclear accumulation of the transcription factor IRF3, and increased expression of *Ifn-β*. Moreover, a chemical inhibitor of cGAS reduced STING-IRF3 signaling and slightly increased multiplication of iKO cultures. Recent work has also implicated cGAS/STING in regulating p21 expression, as depletion of cGAS or STING results in reduced p21 levels and premature mitotic entry [98]. These results support the notion that multiple nuclear and cytoplasmic signaling factors collaborate during the induction and maintenance of the proliferation arrest in ArpC2 iKO cells.

The main mechanism underlying micronucleus biogenesis in Arp2/3-deficient cells is a lack of fidelity in chromatin partitioning during mitosis, as revealed by our live imaging studies. Such segregation defects can be primarily explained by broken DNA fragments failing to properly attach to the microtubule spindle, since most micronuclei formed from cells completing inaccurate mitoses. However, cytoskeletal abnormalities may also contribute to chromosome segregation errors, as changes in the organization of metaphase F-actin and anaphase microtubules are also prevalent in the knockout cells. Our findings complement previous experiments showing altered F-actin levels in meiotic/mitotic structures and defects in spindle formation following chemical Arp2/3 complex inhibition [37–39]. Interestingly, many mitoses in ArpC2 iKO cells feature a prolonged metaphase period. Delayed mitoses arising from failure to satisfy spindle assembly checkpoints can cause chromosome segregation mistakes and premature mitotic exits, both of which give rise to aneuploid cells [99], and both of which were observed at high frequencies in iKO populations. Hence, Arp2/3 complex deficiency leads to multiple deleterious consequences in M-phase of the cell cycle. More work is needed to determine how Arp2/3, actin, and microtubules cooperate in controlling spindle positioning, chromosome alignment, and DNA segregation.

Given that high mutation rates and aneuploidy are associated with organismal aging and tumorigenesis [100], reduced Arp2/3 function *in vivo* could be a contributor to the development of age-related dysfunction and cancers. One central feature of the aging process is an accumulation of senescent cells, which can drive several aging phenotypes [101,102]. Such

discoveries have led to the development of new "senolytic" classes of drugs to treat and slow the progression of age-related pathologies [103]. Our findings suggest that diminished Arp2/3 complex function, and potentially other actin cytoskeletal misregulation, may contribute to premature senescence and aging. Taken together with previous observations that F-actin integrity affects aging and lifespan in *C.elegans* [104,105], preventing or correcting cytoskeletal defects may promote cellular longevity and help reduce the senescent cell burden during organismal aging.

In contrast to the negative impacts of cellular senescence on aging, the anti-proliferative effects of senescence can serve as a positive form of tumor suppression [106]. Several therapeutics that induce senescence in cancer cells have been developed to decrease metastatic growth [107]. The human *CDKN2A* locus, which encodes p16$^{INK4A}$, is frequently inactivated or epigenetically suppressed in various types of cancers [108]. Thus, the *Arpc2* floxed *Cdkn2a* null cells used in our studies present an opportunity to study Arp2/3 complex function in a genetic background that is particularly relevant to preventing the growth of cancer cells. Enhancing our understanding of the connection between the actin cytoskeleton and cellular senescence will therefore provide insight into therapeutic strategies used to regulate cell proliferation, arrest, and death in the context of both age-related diseases and cancers.

## Materials and methods

### Ethics statement

Research with biological materials was approved by the UConn Institutional Biosafety Committee (IBC #58C). This study did not include research with human subjects or live animals.

### ArpC2 Flox and iKO cell culture

MTFs containing the floxed *Arpc2* allele (from James Bear, University of North Carolina) [42] were cultured in DMEM (with 4.5g/L glucose, L-Glutamine, 110mg/L sodium pyruvate), 10% fetal bovine serum (FBS), GlutaMAX, and antibiotic-antimycotic (Gibco). Cells were treated with media containing 0.01% DMSO or 2μM 4-OHT (Sigma) to obtain Flox or iKO populations. For treatments exceeding 3 days, culture supernatants were replaced with fresh media containing DMSO or 4-OHT on day 4. Cultures were returned to normal media after day 6. All experiments were performed using cells that had been in active culture for 2–12 trypsinized passages.

### Mammalian cell lines

NIH3T3 (UC Berkeley cell culture facility), B16-F1 (ATCC), and U2OS (UC Berkeley) cells were cultured in DMEM (with 4.5g/L glucose), 10% FBS, GlutaMAX, and antibiotic-antimycotic. To obtain control or Arp2/3-inhibited populations, culture supernatants were replaced with media containing 0.1% DMSO or 100–200μM CK666 (Millipore) every 12h. Cultures were evaluated for doubling rates, collected for immunoblotting, or fixed for immunofluorescence at 35-60h.

### Mouse fibroblast isolation

PDGFRa-CreER transgenic FVB/NJ mouse tail or ear tissue (from David Goldhamer, UConn) was disinfected in 70% ethanol, air dried, sliced into ~3mm sections, and separated into two 2mL cryovials in 1mL of digestion media (DMEM with 4.5g/L glucose + L-Glutamine + 110mg/L sodium pyruvate supplemented with 12.5mg/mL trypsin and 2.5mg/mL collagenase A) and mixed for 90min at 37°C. The digested contents were homogenized with a 10mL

syringe plunger and then passed through a 70μm cell strainer into 10mL of MTF growth media. Cell suspensions were centrifuged at 580 x g for 7min and cell pellets were resuspended in 10mL of fresh media, re-centrifuged, and resuspended prior to culturing in a 10cm culture dish. MTFs and MEarFs were washed daily until they reached confluency, then trypsinized, passaged, and eventually seeded onto glass coverslips as described below.

## Cell proliferation measurements

MTF, MEarF, NIH3T3, B16-F1, and U2OS cells were cultured in 12-well plates and cell titers were routinely determined using a hemocytometer. Cells were initially seeded at multiple concentrations ranging from $2x10^3$ to $2x10^4$ cells per well. After 5 days, confluent Flox samples were subcultured daily into multiple wells at concentrations of $1-2x10^4$ cells per well, while iKO samples were subcultured if/when they reached 95% confluency. All cultures were expanded into 6-well plates and 6cm dishes when necessary. Population doubling times were calculated based on initial and final cell titers every 24-48h using the equation [time x log(2)] / [log(final)—log(initial)]. Due to their continuous proliferation, the plotted values of cell numbers for Flox samples at days 8–12 were extrapolations based on doubling rates at those time points. For Arp2/3 complex inhibitor experiments, culture supernatants were replaced with media containing DMSO or CK666 every 12h to obtain control or Arp2/3-inhibited populations. Cell titres were counted at 0, 35, and/or 60h. The % of cells in mitosis was quantified based on DAPI staining of condensed chromosomes and confirmed with tubulin staining. For cGAS inhibitor experiments, MTF supernatants were replaced with media containing 5μM RU.521 every 12h from days 4–5.5 after 4-OHT exposure and counted at 0, 24, and 36h of DMSO or RU.521 treatment.

## Flow cytometry

Flox and iKO cells cultured in 10cm dishes were harvested with 1mL of 0.05% Trypsin/EDTA, resuspended in 4mL of media, and centrifuged at 200 x g for 5.5min at 4˚C. Cell pellets were resuspended in 5mL cold phosphate buffered saline (PBS) containing 2% FBS, recentrifuged, and resuspended in 1mL PBS containing 0.5mM EDTA. Cells were then fixed by adding 3.5mL of 100% ethanol during vortexing and placed on ice for 2h. They were then centrifuged at 420 x g for 5.5min at 4˚C, washed with 2mL of PBS, and recentrifuged. Cell pellets were resuspended in 2mL of PBS containing 100μg/mL propidium iodide, 50μg/mL RNaseA, and 0.1% TritonX-100 and incubated in the dark overnight at 4˚C. Flow cytometry measurements were collected using a BD LSRFortessa X-20 Cell Analyzer. FlowJo software (TreeStar, v10) was used to visualize DNA content distribution and the Cell Cycle Analysis tool was used to calculate cell cycle %s.

## DNA transfections and fluorescent probes

For mCherry-cGAS cloning, mouse cGAS was amplified via PCR from a cDNA template (Addgene, 108675) using primers containing KpnI and NotI sites (S1 Table) and inserted into the pKC-mCherryC1 vector [109]. For transfections, Flox cells were grown in 12-well plates for 24h and transfected with 350ng of mCherry-cGAS plasmid using Lipofectamine LTX (Invitrogen) in DMEM. After 3h, DMEM was replaced with MTF media, and 18h later cells were trypsinized and transferred onto 12mm glass coverslips in 24-well plates. Media containing DMSO or 4-OHT was added after 3h, and cells were subjected to additional 29h growth before fixation as described below. For H2B-GFP transfections, Flox cells were grown in 24-well plates for 24h and then transfected with 130ng of H2B-GFP plasmid (Addgene, 11680). After 5h, DMEM was replaced with MTF media containing DMSO or 4-OHT. Cells were imaged

live 15-40h later, as described below. For ArpC2-GFP or GFP-ArpC2 transfections, 4-OHT-treated cells were grown in 24-well plates for 24h and then transfected with 250ng of ArpC2 plasmid (Addgene, 53997 or 53996). 150ng pEGFP-C1 was used as a negative control, and only ArpC2-GFP is shown in S7 Fig. Cells were fixed 29h after transfection. All plasmids were maintained in NEB5-alpha *E.coli* and purified using Macherey-Nagel kits. For imaging acidic cytoplasmic organelles, cells were incubated for 30min in media containing 100nM Lyso-Tracker Red (Invitrogen) prior to fixation.

## RNA interference

For RNAi experiments, cells were grown in 6cm dishes for 24h, treated with DMSO or 4-OHT for 48h, reseeded into 6-well plates, transfected with 40nM siRNAs (S1 Table) using RNAi-MAX (Invitrogen), incubated in growth media for 24h, reseeded into 12-well plates or onto 12mm glass coverslips in 24-well plates, and incubated for an additional 48h. Cells cultured in 12-well plates were counted for proliferation assays, and cells cultured on coverslips were fixed and used in immunofluorescence microscopy assays.

## RT-PCR and RT-qPCR

RNA from Flox and iKO fibroblasts grown in 6-well plates was isolated using TRIzol reagent (Ambion). Following chloroform extraction, isopropanol precipitation, and a 75% ethanol wash, total RNA was resuspended in water. cDNA was reverse transcribed using the iScript cDNA synthesis kit (Bio-Rad) and then PCR-amplified using Taq polymerase (New England Biolabs) and primers listed in S1 Table. Primers were designed to amplify ~340-480bp within each cDNA target. The resulting PCR products were visualized on ethidium bromide-stained agarose gels. For *Ifn-β*, quantification was performed based on band densitometry relative to *Gapdh* using LI-COR Image Studio software. For other targets, RT-qPCR was performed using SYBR-green on a CFX96 Real-Time System (Bio-Rad). 1μl of cDNA was used in each 10μl RT-qPCR reaction, and all samples were run in duplicate. Primer dilution curves were analyzed to ensure primer specificity. Ct values were normalized to GAPDH and/or actin. The iKO:Flox ratio (fold difference) at each timepoint was calculated by the comparative ΔΔ cycle threshold method.

## Cell extracts

For preparation of cell extracts, fibroblasts cultured in 6-well plates were collected in PBS containing 1mM EDTA, centrifuged at 750 x g for 5.5min at 4°C, and lysed in 25mM HEPES (pH 7.4), 100mM NaCl, 1% Triton-X-100, 1mM EDTA, 1mM $Na_3VO_4$, 1mM NaF, 1mM PMSF, and 10μg/ml each of aprotonin, leupeptin, pepstatin, and chymostatin on ice. Lysates were mixed with Laemmli sample buffer, boiled, and centrifuged prior to SDS-PAGE analyses.

## Immunoblotting

Cell extract samples were separated on 12% SDS-PAGE gels before transfer to nitrocellulose membranes (GE Healthcare). Membranes were blocked in PBS containing 5% milk before probing with primary antibodies at concentrations listed in S2 Table. Following overnight incubation at 4°C, membranes were washed and treated with IRDye-680/800- (LI-COR) or horseradish peroxidase-conjugated (GE Healthcare) secondary antibodies. Infrared and chemiluminescent bands were visualized using a LI-COR Odyssey Fc Imaging System. Band intensities were measured using LI-COR Image Studio software. Densitometries of proteins-of-interest were normalized to GAPDH, tubulin, and/or actin loading controls.

## Immunostaining

For immunofluorescence, fibroblasts cultured on glass coverslips in 24-well plates were washed with PBS and fixed using PBS containing 2.5% paraformaldehyde for 30min. Following PBS washes, cells were permeabilized using PBS containing 0.1% Triton X-100 for 2min, washed, and stained with primary antibodies in PBS containing 1% bovine serum albumin, 1% FBS, and 0.02% azide for 45-60min as described in S2 Table. Cells were washed and treated with Alexa Fluor 488-, 555-, or 647-conjugated goat anti-rabbit, anti-mouse, or anti-rat secondary antibodies, 1μg/mL DAPI, and/or 0.2U/mL Alexa Fluor 488- or 647-labeled phalloidin (Invitrogen) for 35-45min as detailed in S2 Table. Following washes, coverslips were mounted on glass slides in ProLong Gold anti-fade (Invitrogen).

## Fluorescence microscopy

All fixed and live cells were imaged using a Nikon Eclipse Ti microscope equipped with Plan Apo 100X (1.45 NA), Plan Apo 60X (1.40 NA), or Plan Fluor 20X (0.5 NA) objectives, an Andor Clara-E camera, and a computer running NIS Elements Software. Most images were taken as single epifluorescence slices, whereas 60X images of mCherry-transfected cells (Fig 10) were taken as z-stacks with a 0.3μm step size, and fixed mitotic cells were taken with a 0.3μm (Fig 6) or 0.5μm (Figs 5 and 7) step size. Live cell imaging was performed in a 35˚C chamber (Okolab). During live imaging, cells were cultured in fresh media containing 25mM HEPES (pH 7.4) and DMSO or 4-OHT. Images were captured using the 20x objective at 12min intervals.

## Fluorescence quantification

All image processing and analysis was conducted using ImageJ/FIJI software [110]. Coverslips were coded alphanumerically prior to scoring. For fibroblast area calculations, cell perimeters were outlined based on phalloidin staining. The distributions of MTF cell sizes depicted in Fig 1I are reflected in the heterogeneity of cell sizes presented in subsequent figures. For Lamin B1 abundance, the mean fluorescence intensity (MFI) of Lamin B1 staining in the DAPI-stained nuclear area of the cell was measured and the background Lamin B1 signal from outside the cell was subtracted from the MFI values. For normalization purposes, all corrected MFI values of individual Flox and iKO cells were divided by the overall average MFI across 2 experiments for Flox cells. For acidic organelle analyses, the number of cells with diffuse Lyso-Tracker staining was scored as positive or negative and divided by the total number of cells analyzed. To determine the LysoTracker area as a % of the total cell area, cell perimeters were outlined based on phalloidin staining and the Threshold tool was used to quantify the % of this area containing LysoTracker staining. For micronuclei and γH2AX quantifications, the number of cells with DAPI-stained micronuclei and the number of cells with prominent nuclear γH2AX clusters were quantified manually and divided by the total number of cells analyzed. Clusters were defined as isolated and discrete regions of γH2AX staining. For analyses of nuclear γH2AX intensity, the MFI of γH2AX staining in the DAPI-stained nuclear area of the cell was measured and the background signal from outside the cell was subtracted from the MFI values. The average corrected MFI of DMSO-treated cells was set to 1. For analyses of p21, the MFI of p21 staining in the DAPI-stained nuclear area of the cell was measured and the average p21 MFI across 3 experiments for Flox and iKO cells was calculated. The average MFI of Flox cells was set to 1. In RNAi experiments, the MFI of p21 staining in the nuclear area was measured and the lowest nuclear p21 signal was subtracted from the MFI values before the MFI values were normalized to DAPI MFI values. Nuclear P-p53 and IRF3 intensities were measured similarly but relied on 60x rather than 20x images. The ratio of nuclear-to-

cytoplasmic IRF3 fluorescence intensity was quantified using 5μm diameter circles drawn in the nucleus and cytoplasm. For area-based assays of F-actin and microtubule intensities during metaphase, the DAPI-stained DNA mass (Fig 6) or 5μm diameter circles around the centrosomal spindle poles (S8 Fig) were outlined, and the MFIs of phalloidin and anti-tubulin staining in these areas were measured relative to DAPI. For linescan analyses of F-actin, tubulin, ArpC2, and DNA staining intensities in metaphase and/or anaphase cells, the Plot-Profile tool was used. For metaphase, 14–16μm lines were drawn across the chromatin mass. For anaphase, 10–15μm lines were drawn across the mitotic spindle halfway between one set of chromosomes and the equator of the two forming daughter cells. In each plot, the minimum pixel intensity recorded along the line was subtracted from all values along the line to set the minimum to 0, then all values were divided by the maximum to set the highest peak to 1. For measuring the left:right tubulin intensity ratios of anaphase cells, the two halves of the spindle were outlined and the MFI was measured. The half with the highest intensity was called left. The numbers of cells analyzed in each type of assay are listed in the Figure Legends.

### β-galactosidase assays

SA-βgal activity was assessed using the Senescence β-Galactosidase Staining Kit (Cell Signaling Technologies, 9860). Fibroblasts were cultured in 6-well plates, washed once with PBS, fixed for 20min, and washed twice with PBS before incubation in SA-βgal staining solution at 37˚C in the dark for 20-24h. Images were captured using an iPhone 7 on a bright-field microscope equipped with a 10x objective. The % of SA-βgal-positive cells was quantified by counting the number of intensely blue-colored cells and dividing by the total number of cells. Samples were coded and scored in a blinded fashion.

### Reproducibility and statistics

All conclusions were based on observations made from at least 4 separate experiments, while quantifications were based on data from 2–6 representative experiments. Statistical analyses were performed using GraphPad Prism software as noted in the Figure Legends. P-values for data sets including 2 conditions were determined using unpaired t-tests. Analyses of data sets involving +/- scoring pooled from multiple experiments (e.g., Figs 5C–5E and 11C-11D) used Fisher's exact test. P-values $<0.05$ were considered statistically significant.

### Supporting information

**S1 Table. Primers.**
(PDF)

**S2 Table. Immunofluorescence and Immunoblotting Reagents.**
(PDF)

**S1 Fig. ArpC2 iKO cells stall in the G1 phase of the cell cycle. (A)** *Arpc2*-floxed mouse tail fibroblasts (MTFs) were treated with DMSO (Flox) or 4-OHT (iKO) for 6d, transferred to a 6cm dish containing drug-free media, and examined by light microscopy using an iPhone at 12d. Arrowheads (left panel) and circular outlines (right panel) highlight colonies that either avoided CreER-mediated recombination at the *Arpc2* locus, converted to a functional *Arpc2*-expressing variant, or acquired suppressors of the *Arpc2* deletion. Colony formation was observed for approximately 1 out of every 100,000 cells. **(B)** *Arpc2*-floxed MTFs were treated with DMSO (Flox) or 4-OHT (iKO) for 6d, collected, fixed, stained with propidium iodide, and analyzed by flow cytometry. 10,000 events were examined for each cell type. The % of cells

in each phase of the cell cycle was quantified using FlowJo software.
(TIF)

**S2 Fig. Pro-IL-1β is expressed at similar levels in ArpC2 Flox and iKO cells.** Mouse fibro-blasts were treated with DMSO (Flox) or 4-OHT (iKO) for 6d and collected at 9d. Samples were lysed, subjected to SDS-PAGE, and immunoblotted with antibodies to IL-1β, tubulin, actin, and GAPDH. Pro-IL-1β densitometry values are shown. Cleaved IL-1β was not detected.
(TIF)

**S3 Fig. 4-OHT-mediated induction of CreER does not cause cell cycle arrest, trigger micro-nucleus biogenesis, or increase dsDNA breaks. (A)** Mouse tail fibroblasts (MTFs) with Plate-let Derived Growth Factor Receptor alpha promoter-driven expression of a Cre recombinase fused to a human estrogen receptor ligand binding domain (PDGFRa-CreER) were treated with DMSO or 4-OHT for 3d, fixed, and stained with phalloidin (F-actin; magenta), a γH2AX antibody (green), and DAPI (DNA; blue). Arrowheads point to micronuclei. **(B)** PDGFRa-CreER MTFs or PDGFRa-CreER mouse ear fibroblasts (MEarFs) were treated with DMSO or 4-OHT for 6d, and population doubling times were quantified from 8-10d. Each bar represents the mean doubling time ±SD from n = 3 experiments. 4-OHT-treated cells replicated faster, not slower, than DMSO-treated control cells. **(C)** The % of PDGFRa-CreER MTFs with micro-nuclei was quantified following a 3d exposure to DMSO or 4-OHT. Each bar represents the mean % ±SD from n = 3 experiments (812–899 cells per bar). **(D)** Nuclear γH2AX levels were quantified by outlining the DAPI-stained nucleus of each cell in ImageJ and measuring the mean γH2AX pixel intensity. Each bar represents the mean % ±SD from n = 3 experiments. RFU = Relative Fluorescence Units.
(TIF)

**S4 Fig. Prominent DNA damage clusters are found in the nuclei and micronuclei of ArpC2 iKO cells.** Mouse fibroblasts were treated with DMSO (Flox) or 4-OHT (iKO) for 6d, fixed at 7d, and stained with a γH2AX antibody (green) and DAPI (DNA; blue). Arrowheads point to γH2AX clusters in micronuclei and arrows indicate γH2AX clusters in nuclei.
(TIF)

**S5 Fig. Pharmacological inhibition of the Arp2/3 complex impairs cell proliferation, increases micronucleus biogenesis, and increases dsDNA breaks in other mouse cell lines. (A)** NIH3T3 and B16-F1 cells were treated with DMSO or 100μM CK666 for 35h (with DMSO or CK666 media changes at 0h, 12h, and 24h), fixed, and stained with phalloidin (F-actin; magenta), an anti-tubulin antibody (green), and DAPI (DNA; blue). Only NIH3T3 cells are shown. Arrowheads highlight micronuclei. **(B)** The % of NIH3T3 cells in mitosis was quantified for samples treated as in A. Each bar represents the mean % ±SD from n = 3 experi-ments (999–1684 cells per bar). **(C)** NIH3T3 and B16-F1 cells were treated as in A and popula-tion doubling times were quantified. Each bar represents the mean doubling time ±SD from n = 3 experiments. **(D)** NIH3T3 and B16-F1 cells were treated as in A and the % of cells with micronuclei was quantified. Each bar represents the mean % ±SD from n = 3 or 4 experiments (354–602 cells per bar). **(E)** NIH3T3 and B16-F1 cells were treated with DMSO or CK666, fixed, and stained with a γH2AX antibody (green) and DAPI. **(F)** Nuclear γH2AX levels were quantified by outlining the DAPI-stained nucleus of each cell in ImageJ and measuring the mean γH2AX pixel intensity. Each bar represents the mean % ±SD from n = 3 or 4 experi-ments (100–240 cells per bar). RFU = Relative Fluorescence Units.
(TIF)

**S6 Fig. Pharmacological inhibition of the Arp2/3 complex impairs cell proliferation and increases dsDNA breaks in human U2OS cells. (A)** U2OS osteosarcoma cells were treated with DMSO or 200μM CK666 for 60h (with DMSO or CK666 media changes at 0h, 12h, 24h, 36h, and 48h), fixed, and stained with phalloidin (F-actin; magenta) and DAPI (DNA; blue). **(B)** The % of U2OS cells in mitosis were quantified for samples treated as in A. Each bar represents the mean % ±SD from n = 3 experiments (1198–1454 cells per bar). **(C)** Cells were treated as in A and population doubling times were quantified. Each bar represents the doubling time from a representative experiment. **(D)** Cells were treated as in A and the % of cells with micronuclei was quantified. Each bar represents the mean % ±SD from n = 3 experiments (1198–1454 cells per bar). Surprisingly, CK666-treated cells had fewer micronuclei than DMSO-treated cells. **(E)** Cells were treated with DMSO or CK666, fixed, and stained with a γH2AX antibody (green) and DAPI. **(F)** Nuclear γH2AX levels were quantified by outlining the DAPI-stained nucleus of each cell in ImageJ and measuring the mean γH2AX pixel intensity. Each bar represents the mean % ±SD from n = 3 experiments (890–937 cells per bar). RFU = Relative Fluorescence Units.
(TIF)

**S7 Fig. Expression of ArpC2-GFP prevents γH2AX accumulation in ArpC2 iKO cells. (A)** Mouse fibroblasts were treated with 4-OHT (iKO), transfected with plasmids encoding GFP or ArpC2-GFP (green), fixed at 2d, and stained a γH2AX antibody (magenta) and DAPI (DNA; blue). Arrowheads point to micronuclei. **(B)** Nuclear γH2AX levels were quantified by outlining the DAPI-stained nucleus of each cell in ImageJ and measuring the mean γH2AX pixel intensity. Each bar represents the mean % ±SD from n = 12–24 cells per category. RFU = Relative Fluorescence Units. **(C)** Cells were transfected as in A and stained with phalloidin (F-actin; magenta). An ArpC2-GFP-expressing cell in metaphase is shown. Note the penetration of ArpC2-GFP and F-actin into the central chromatin mass.
(TIF)

**S8 Fig. Metaphase chromatin-associated but not centrosome-associated F-actin levels are reduced in ArpC2 iKO cells. (A)** Mouse fibroblasts (Flox) were treated with DMSO for 1-2d, fixed, and stained with phalloidin (F-actin), anti-ArpC2 antibodies (yellow), an anti-Arp3 antibody (cyan), and DAPI (DNA; blue) as in Fig 6A. **(B)** The DNA-containing region was isolated from representative spindles in Fig 6B and 6C and magnified. **(C)** Mouse fibroblasts were treated with DMSO for 1-2d, fixed, and stained with phalloidin, an anti-tubulin antibody, and DAPI as in Fig 6C. Circles of 5μm diameter were drawn around centrosomes in metaphase cells and the fluorescence intensities of F-actin and microtubules were measured as in Fig 6D. Each bar represents the mean intensity ±SD from n = 24 metaphase centrosome-associated regions compiled from 3 experiments.
(TIF)

**S9 Fig. Expression of mCherry-cGAS in ArpC2 iKO cells.** Mouse fibroblasts were treated with 4-OHT (iKO), transfected with plasmids encoding mCherry or mCherry-cGAS as in Fig 10A, collected, and immunoblotted with antibodies to mCherry and GAPDH.
(TIF)

**S10 Fig. Model: Pathway to senescence resulting from Arp2/3 complex deletion.**
(TIF)

**S1 Dataset. File containing the data underlying the summary graphs.**
(XLSX)

**S1 Video. Timelapse movie of a H2B-GFP-expressing Flox cell (see Fig 5A first row).** (AVI)

**S2 Video. Timelapse movie of a H2B-GFP-expressing Flox cell (see Fig 5A second row).** (AVI)

**S3 Video. Timelapse movie of a H2B-GFP-expressing iKO cell (see Fig 5A third row).** (AVI)

**S4 Video. Timelapse movie of a H2B-GFP-expressing iKO cell (see Fig 5A fourth row).** (AVI)

**S5 Video. Timelapse movie of a H2B-GFP-expressing iKO cell (see Fig 5B first row).** (AVI)

**S6 Video. Timelapse movie of a H2B-GFP-expressing iKO cell (see Fig 5B second row).** (AVI)

**S7 Video. Timelapse movie of a H2B-GFP-expressing iKO cell (see Fig 5B third row).** (AVI)

## Acknowledgments

We thank Jim Bear (UNC Chapel Hill) for providing cells harboring the floxed *ArpC2* allele, Amanda Harrop and David Goldhamer (UConn) for providing *Pdgfra-CreER* mouse tails and ears, Wu He (UConn Flow Cytometry Facility) for assistance with FACS analyses, Ming Xu (UConn Health) for comments on senescence experiments, Tom Maresca (UMass Amherst) for input on mitosis experiments, Tom Perrotta for support with writing, and Campellone Lab members for their suggestions related to this manuscript. ELH was supported by the UConn Office of Undergraduate Research, SG was supported by the UConn McNair Scholars Program, and RBF was supported by the UConn Holster Scholars Program.

## Author Contributions

**Conceptualization:** Elena L. Haarer, Kenneth G. Campellone.

**Data curation:** Elena L. Haarer, Shirley Guo, Kenneth G. Campellone.

**Formal analysis:** Elena L. Haarer, Corey J. Theodore, Shirley Guo, Ryan B. Frier, Kenneth G. Campellone.

**Funding acquisition:** Kenneth G. Campellone.

**Investigation:** Elena L. Haarer, Corey J. Theodore, Shirley Guo, Ryan B. Frier, Kenneth G. Campellone.

**Methodology:** Elena L. Haarer, Corey J. Theodore, Shirley Guo, Ryan B. Frier, Kenneth G. Campellone.

**Project administration:** Kenneth G. Campellone.

**Resources:** Kenneth G. Campellone.

**Software:** Ryan B. Frier.

**Supervision:** Kenneth G. Campellone.

**Validation:** Elena L. Haarer, Corey J. Theodore, Shirley Guo, Ryan B. Frier, Kenneth G. Campellone.

**Visualization:** Elena L. Haarer, Corey J. Theodore, Shirley Guo, Ryan B. Frier, Kenneth G. Campellone.

**Writing – original draft:** Elena L. Haarer, Kenneth G. Campellone.

**Writing – review & editing:** Elena L. Haarer, Corey J. Theodore, Shirley Guo, Ryan B. Frier, Kenneth G. Campellone.

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
