## [Decision Letter · Decision Letter 0]

11 Mar 2022

Dear Dr Campellone,

Thank you very much for submitting your Research Article entitled 'Genomic instability caused by Arp2/3 complex inactivation results in micronucleus biogenesis and cellular senescence' to PLOS Genetics.

The manuscript was fully evaluated at the editorial level and by three independent peer reviewers. The reviewers appreciated the attention to an important problem, but raised some substantial concerns about the current manuscript. Based on the reviews, we will not be able to accept this version of the manuscript, but we would be willing to review a much-revised version. We anticipate that the revision will require additional experimentation. We cannot, of course, promise publication at that time.

If you decide to revise the manuscript for further consideration at PLOS Genetics, please aim to resubmit within the next 60 days, unless it will take extra time to address the concerns of the reviewers, in which case we would appreciate an expected resubmission date by email to plosgenetics@plos.org.

[LINK]

We are sorry that we cannot be more positive about your manuscript at this stage. Please do not hesitate to contact us if you have any concerns or questions.

Yours sincerely,

Gregory P. Copenhaver, Ph.D.

Editor-in-Chief

PLOS Genetics

Gregory Barsh

Editor-in-Chief

PLOS Genetics

Reviewer's Responses to Questions

**Comments to the Authors:**

Reviewer #1: Elena L. Haare et al. showed in this manuscript that ArpC2 depletion induced cell cycle arrest with up-regulation of senescence markers such as p21/p53, DNA damage markers, and SAbetaGal staining in p16 knockout mouse tail fibroblasts. They also found that ArpC2 depletion resulted in the cGAS STING recruitment to the micronuclei and the formation of cytosolic micronuclei. These findings about the association between cytoskeletal dysfunction and cellular senescence are interesting. Using the several senescent markers and assays, the authors clearly showed that ArpC2 iKO cells became senescence like phenotype, however I have some concern about the molecular mechanisms and causality. Especially, the mechanisms about cGAS STING pathway and micronuclei formation are descriptive so far. Furthermore, they did all experiments in p16 knockout mouse tail fibroblasts. The following issues should be addressed to make the findings more robust and conclusive.

Major issue

1. In Fig 1A, why Arp3 expression was decreased in ArpC2 iKO cells? To confirm the specificity of ArpC2 iKO model, the protein and mRNA expression level of all seven subunits of Arp2/3 complex should be shown. The word in the title “Arp2/3 complex inactivation” is also obscure, not accurate because they only deplete ArpC2 and provide only ArpC2 and Arp3 expression.

In addition, it was reported that single depletion of either ArpC2 or Arp2 in the Ink4a/Arf-/- IA32 cell line was not stable and the knockdown of both subunits were needed to produce a stable Arp2/3 knockdown (Cell. 2012 Mar 2;148(5):973-87.). Although ArpC2 iKO is different system from shRNA targeting ArpC2, the data which is not consistent with the previous data needs careful verification and mechanisms.

2. All the experiments showed in this manuscript are done with p16 knockout mouse tail fibroblasts. The author mentioned about the previous study in line 117 of page 7, which showed knockdown of both the ArpC2 and Arp2 subunits are culturable in a p16Ink4a/Arf Ko cells, however in this paper the growth curve of Arp2 shRNA in wild-type and Arf-/- MEFs were showed (Cell. 2012 Mar 2;148(5):973-87.). They also described in the discussion that “ArpC2 iKO induce p16 independent cellular senescence”. They should try to show the senescence induction by knockdown of Arp2/3 using other tumor cell lines. If it works, it is more interesting to induce senescence to tumor cell.

3. In 154 of page 8, “apoptotic cell phenotypes were not observed in the iKO cell population”.

If they don’t provide the data, this should be omitted.

4. In Fig 2A and 2B, showing only IL-6 is not enough for the evidence of increase of SASP. IL-1beta, IL-1 alpha, TNF alpha, IL-8, IL-10, and GDF15 etc should be measured.

5. In Fig 6A, 7C, WB data should be with quantification data.

6. In Fig 7, the molecular mechanisms related with cGAS/STING signal and p53/p21 signals are descriptive. How cytoskeletal dysfunction activate cGAS/STING signals? The immunostaining data were not reliable at all. TBK1, NfkB, IRF3, and INFb expression etc should be analyzed.

7. In Fig 7F, validation data of RU.521 such as inhibition of downstream signals (NfkB, IRF3, and INFbeta) should be provided.

Line 311 in page 14, If they mention “that cGAS inhibition has the potential to oppose the initiation of senescence in some of the cells in this experimental system. Collectively, the above localization and inhibitor results support the idea that cGAS/STING signaling contributes to the establishment of senescence in Arp2/3-deficient cells.”, they should show the key data in Fig1-4, 6 using RU.521.

8. To show the robust causality, the “rescue” experiment, overexpression of ArpC2 in iKO cells should be considered.

9. Line 478 in page 21, to show that “ArpC2 iKO cells activate p53 and accumulate p21 in a p16- independent manner”, the data using p53 inhibitor, commercially available, or using p53 mutated cell lines would be important.

10. Are the expression of each Arp2/3 subunits increased in senescent cells?

11. Regarding “impairment of Arp2/3 function in vivo could be a contributor to the development of age-related dysfunction and cancers” in line 511 of page 22, is there any evidence that the relationship between Arp2/3 mutation or SNPs and diseases?

In this manuscript, they showed only in vitro data using only p16 knockout mouse tail fibroblasts. That's a huge leap in logic.

Reviewer #2: The study by Haarer et al. focuses on the role of Arp2/3 during mitosis and the associated genomic integrity. The study is based on an elegant method for an inducible knock out (KO) of the ArpC2 subunit. The ArpC2 KO resulted in DNA damage, formation of cytosolic micronuclei and senescence. In the last part of the study the authors analyze changes in the organization of mitotic spindle actin and spindle microtubules.

As the authors mention in their discussion spindle defects can cause chromosome segregation mistakes and premature mitotic exits which can explain most of the phenotypes observed in ArpC2 KO cells (i.e. DNA damage, micronuclei and senescence). Indeed, previous studies using Arp2/3 inhibitors have revealed spindle- and chromosome segregation defects. It could be a point of criticism that the results are somehow expected from previous studies. However, the study is experimentally solid and gives a broad scope for the importance of Arp2/3 in mitosis and associated genomic stability. The focus on mitosis, DNA damage and senescence sets this study apart from many previous studies.

Despite the sound experimental layout additional experiments and changes in the presentation should be made to increase the significance of the study.

Reviewer questions / suggestions for experiments and data presentation:

1.) In Figure 1F an increase in cell area is shown and quantified in 1G. The difference is very robust in the field of view and quantification. This difference cannot be seen in Fig. 1C. Both images t=7d. Has Flox a different scale bar that was forgotten?

The very uniform increase in cell- and nuclear area in Fig.1F (t=d7) is very heterogeneous in Fig. 2E (t=d9). Why?

2.) In Figure 1F and G a clear increase in cell area and nuclear area is shown. The difference is so obvious that a flatter cell and nucleus are probably not the reason. Due to nuclear cytoplasmic scaling, this reviewer assumes that the cells are all already multiploid. This could be checked by a volume analysis and/ or FACS analysis for the DNA content (over time would be very good). This is interesting because here it looks like all cells (t=7d) had the same fate as the cells in Fig. 8B with premature mitotic exit. This could be the main phenotype leading to others like a senescent state. If this is true, it should be discussed. What is the ratio of cells with full KO t>5d complete mitosis vs. premature mitotic exit for example?

3.) In Fig. 2E magnifications should be provided that better show the phenotype to the reader.

4.) The graphical presentation of Fig. 2F could be improved. Individual datapoints seem to cover and hide each other. This can be avoided for superblots in the latest versions of programs like prism.

5.) In Fig. 3C+D images with higher resolution would help the reader. In Flox distinct lysosomes are visible. In KO the staining looks diffuse. However, it is likely a strong accumulation of lysosomes that just looks like a diffuse mass due to low resolution.

6.) Fig. 4A far right cell (KO) with micronuclei seems to be Fig.4B (vi) but mirrored. This reviewer has no issues with reusing this example in B. However, the mirror effect should be removed. If the authors agree, a sentence could tell the reader: vi is from Fig. 4A. If the authors see B as a fully independent figure, they might add to the legend of 4B: DNA was stained with DAPI.

In general figure legends are rather short and could be revised that all information is given. As an alternative, phrases like “treated as in A” could be used.

7.) In Fig. 6E seems to be a correlation between p21 intensity and nuclear size (multiploidy by mitosis defects? Please see point 2.) Is this true? Would it be interesting to analyze or discuss this?

8.) In Fig. 7D+E the authors mention a speckled ER-like localization of STING. In Fig. 7E this looks very randomly cytoplasmic. However, after zooming in, in Fig.7D the ER-localization seems to be nice and crisp. It appears feasible that the authors could provide a much better example in Fig. 7E.

9.) In Fig. 8 the authors show nice mitosis defects. The figure layout appears narrow enough to add a magnification of the defect at the final timepoint. The defects were only visible after excessive zooming. A magnification will help the reader a lot going through the figures.

Related to that figure the authors provide video files. The submitted video files have a very poor quality compared to the still images in Fig. 8A+B. Can this problem be addressed by a better video conversion?

10.) Fig. 9B shows quite nice the penetration of actin and microtubules in the metaphase plate. It would be helpful for the reader to have the same for KO cells.

In the low mag. images Fig. 9C it looks like the DNA containing region is more devoid of actin filaments and microtubule bundles. Quantification 9D shows that for actin but not significant for tubulin. A bulk fluorescence measurement in the chromatin area is not very robust for analyzing a complex structure like the spindle. However, the line scan 9E shows less prominent peaks for tubulin (microtubule bundles) than in Flox cells. The same is true for Fig. 10A+B. Also, the even microtubule distribution (symmetry) is affected as the authors mention.

Arp-dependent actin polymerization seems to be necessary for proper microtubule spindle organization. Is this a take home message? Should it be discussed in more detail?

11.) The authors suggest that “in addition to the presence of misplaced damaged chromatin and a decrease in actin filaments at the metaphase plate, chromosome missegregations arising from alterations in anaphase microtubule organization MAY BE A CONTRIBUTING FACTOR in the formation of micronuclei in Arp2/3-deficient cells”. Most micronuclei form during mitosis. ArpC2 KO causes spindle defects. This could lead to the hypothesis that Arp-dependent actin polymerization is important for spindle organization, KO of Arp causes spindle defects and mitosis errors which are the MAIN REASON for micronuclei, DNA-damage and senescence in this experimental setup. If mitosis is blocked / the cell cycle arrested… do KO cells accumulate DNA damage to a similar extent? If the block is released, are they senescent or is senescence delayed until damage is acquired by additional mitosis?

12.) Some experiments (without time course, most have a time course) were performed at different time points after induction of the KO. The authors should carefully clarify for the reader what was the ratio behind choosing different time points.

Moreover, were certain phenotypes only visible at certain timepoints? For example, complete mitosis yielding micronuclei is a phenotype connected to partial depleted Arp functionality. Premature mitotic exit is prevailing at stronger Arp depletion. Are spindle defects more severe at d3-4 than on d1? The advantage of the system is that the authors can nicely correlate phenotypes to certain levels of depletion down to a full KO.

13.) There are established ways to inhibit Arp2/3 functionality (CK-666, Arpin…). The reviewer is aware of the elegant system the authors use…Using a different way of inhibiting Arp2/3 functionality in 1 or 2 crucial experiments could further rule out the possibility of KO off target effects or effects by 4-OHT.

Also 1-2 crucial experiments with an ArpC2 rescue by reexpression at an early timepoint, would be a valuable control of the results.

Reviewer #3: In this work, the authors study effects of Arp2/3 complex inactivation on micronucleus biogenesis and the induction of cellular senescence. This is an exciting and very important topic, but unfortunately, the authors have chosen not to dig very deeply into the precise mechanistic reasons for the observations made. This also becomes clear when reading the discussion. I agree that the reasons for how Arp2/3 complex impacts on the these processes could be multiple, but this has already been evident before reading the article, so the question remains what precisely we can learn from the present study. It also seems a bit as if the authors have tried to pull on several ends simultaneously and have then thrown everything into one paper, but they failed, unfortunately, to come up with a conclusive story and key observations. Just as one example: the changes in lysotracker staining upon induced ArpC3-KO shown in Figure 3C are really dramatic, but what does this mean and how are these observations explained? As opposed to looking into this in more detail, the authors just moved on to the next observation, which makes the study quite descriptive. I do like the evidence that the micronuclei found in about 25% of the cells upon KO form during mitosis (and not other stages of the cell cycle), this is nicely demonstrated using video microscopy (Fig. 8), but the mechanistic reason for this is missing. I really have problems to believe in the described changes in actin filament penetration into the central spindle upon Arp2/3-KO or the described asymmetries in microtubule spindle organization (Figure 10). There are no convincing quantifications for the latter, and the data seem quite superficial. At this stage, I have just listed a few key points and problems that have come to my mind when reading this study.

1) Figure 1 is very thorough and nice, and should thus definitely been shown as is, although an increase in cell size upon acute Arp2/3 elimination was previously described in the literature (see e.g. PMID: 33598464), so this should at least be mentioned.

Aside from this though, I wondered: have the authors tried to clone out cell lines that escape from this suppression of proliferation? Importantly, a recent study described the generation of an HL60 cell line stably lacking Arp2 (PMID: 31600188, also used in 34096975), and I was disturbed by the fact that the authors did not even mention these recent studies published in in PLoS Biol and JCB. Later in the current work, the authors described genomic problems such as the formation of micronuclei in just a subfraction of cells (20% in Fig. 4C), so does this mean that cells not developing this phenotype could be convinced to continue proliferating, perhaps with conditioned media from wildtype cells? The authors are actually discussing paracrine phenomena as potential mechanisms in this context (see Discussion), so why was this not experimentally addressed?

2) The observation in Fig 2 on the reduction in lamin B1 levels is very interesting, and likely important. It is also completely novel to my knowledge, but this is where the study ends, which is a shame. It would have been very interesting to address what the consequences of these reduced lamin B1 levels are – lamins have been previously connected to the development of progeria, correct? And it would have also been interesting to study here where this reduction in lamin B1 comes from – what is the mechanism of lamin B1 suppression? Can this be circumvented by ectopic expression? Can phenotypes of Arp2/3 KO described below be rescued by restoring lamin B1 levels?

3) The observations in Figure 3 are also interesting, but again, no follow-up experiments are performed. For instance, the changes in lysosome staining are stunning (Fig. 3C) and the consequences likely dramatic potentially, but without additional experiments, the results sort of stand alone without much context concerning specific Arp2/3 complex functions, which makes the study quite descriptive in places.

4) In general, Figures 2-5 display quite small datasets all together, so could be easily combined in one way or the other.

5) The cGAS and STING stainings shown in Fig 7 are not as relevant in my view as some of the other observations, and could be easily moved to the Supplement perhaps. I also don’t share the authors’ view on the cGAS inhibitor experiments (RU.521). The conclusion by the authors was that RU.521 reduces the population doubling time upon Arp2/3 KO in a statistically significant fashion (Fig. 7F), but the effect was so tiny that I almost couldn’t see it initially. I also wonder what sort of statistics was used for comparing these datasets, and how such minute changes could provide statistically significant differences (p-value of 0.02). I think looking at the Figure superficially, the conclusion could have just as well easily been that RU.521 does NOT show much of an effect at all.

6) Figure 8 makes sense, but I was much less convinced with the observations described in Figs. 9 and 10. I really cannot discern a convincing, specific ArpC2-staining on or in the metaphase plate in Fig. 9A, so the statement (lines 380/1) that “these observations expand the catalog of F-actin and Arp2/3-associated cytoskeletal structures that are found within dividing mammalian cells” is certainly an over-statement. What about performing live-cell, confocal imaging of cells expressing EGFP-tagged Arp2/3 complex subunits? How about other Arp2/3 complex subunit antibodies? What do they show? There is no reason to restrict these stainings to just ArpC2. Are the authors sure the antibody is functional at all in IF? Can this be confirmed in interphase, control cells displaying lamellipodia? In addition, in Figure 9D, the quantification is concluded to show that actin filament intensities are reduced (which is perhaps the case), but microtubules are not. The reason for the latter conclusion appears to be the lack of statistical significance, but a trend of reduction is certainly observable as well, so here the drawn conclusions also seem to diverge quite significantly from the data shown (as above). If it holds true that removal of Arp2/3 reduces actin filaments in the metaphase plate during cell division, this should be studied and demonstrated using a variety of methods (including live cell, confocal imaging) much more thoroughly, because this then could be quite important, but the data as currently presented don’t show this in a convincing fashion.

7) The potential misalignment of spindle microtubules in anaphase could be interesting as well (Figure 10), but again, the authors just show a very preliminary first dataset (basically one cell in each experimental group), without quantification how frequently this would occur, so again, the experiments sort of stop half way in between. Here, the question arises again where the potential mis-alignment comes from precisely? I guess it is difficult to exclude at least that this might also have to do with problems with centrosome function upon Arp2/3 depletion described by others previously, which is a possibility not even mentioned here. Although I am aware of the notion that spindles can self-assemble in principle in the absence of centrosomes, I am not sure whether one can exclude an impact of changes of actin assembly at centrosomes (caused by Arp2/3-KO) on the organization and dynamics of spindle microtubules, in particular in later stages of mitosis. Maybe I am wrong here and if this can really be excluded, but then it would be important to discuss it at least, and to perform more direct experiments to demonstrate how interference with Arp2/3 complex function during cell division mechanistically causes anaphase microtubule asymmetry and/or modified/abrograted spindle microtubule patterns and dynamics.

**Have all data underlying the figures and results presented in the manuscript been provided?**

Reviewer #1: Yes

Reviewer #2: Yes

Reviewer #3: Yes

PLOS authors have the option to publish the peer review history of their article (what does this mean?). If published, this will include your full peer review and any attached files.

Reviewer #1: No

Reviewer #2: No

Reviewer #3: No

---

## [Decision Letter · Decision Letter 1]

13 Dec 2022

Dear Dr Campellone,

Thank you very much for submitting your Research Article entitled 'Genomic instability caused by Arp2/3 complex inactivation results in micronucleus biogenesis and cellular senescence' to PLOS Genetics.

The manuscript was fully evaluated at the editorial level and by one of the original independent peer reviewers. The reviewer appreciated that you have made strides in revealing mechanistic insight, but have concerns that for these insights to be robust they need to be supported by quantitative analysis. Based on the reviews, we will not be able to accept this version of the manuscript, but we would be willing to review a much-revised version. We cannot, of course, promise publication at that time.

If you decide to revise the manuscript for further consideration at PLOS Genetics, please aim to resubmit within the next 60 days, unless it will take extra time to address the concerns of the reviewers, in which case we would appreciate an expected resubmission date by email to plosgenetics@plos.org.

We are sorry that we cannot be more positive about your manuscript at this stage. Please do not hesitate to contact us if you have any concerns or questions.

Yours sincerely,

Gregory P. Copenhaver

Editor-in-Chief

PLOS Genetics

Gregory Barsh

Editor-in-Chief

PLOS Genetics

Reviewer's Responses to Questions

**Comments to the Authors:**

Reviewer #1: Elena L. Haare et al. showed in this manuscript that ArpC2 depletion induced cell cycle arrest with up-regulation of senescence markers such as p21/p53, DNA damage markers, and SAbetaGal staining.

Some molecular mechanisms were shown in the revision manuscript. p21 senescent pathway was involved in the cytoskeletal dysfunction induced cellular senescence, and IRF3 high cells were significantly decreased by cGAS/STING signal inhibition. But quantification data of IRF3 >3 fold high cells are very subjective. Most of the data were shown as immunostaining data. Immunostaining is not useful tool for quantification in general. It is better to show a combination of other assays.

In addition, as other reviewer mentioned, they showed that the cell area was increased in iKO cells, however, in Fig 1C, Fig 3C, Cig 8C, Fig 10D, and Fig 11B, cell size in iKO cells looks same as Flox cells.

I propose that the data about senescence markers should be in one figure, because a combination of assays is needed to determine senescent cells. For example, cell cycle arrest (Fig 2D-G), beta gal (3A, B), p53(Fig8A-C), p21(Fig9A-C), LaminB1(Fig2E, F), and SASP markers (Fig2A-D).

**Have all data underlying the figures and results presented in the manuscript been provided?**

Reviewer #1: Yes

PLOS authors have the option to publish the peer review history of their article (what does this mean?). If published, this will include your full peer review and any attached files.

Reviewer #1: No

---

## [Editor Report · Decision Letter 2]

10 Jan 2023

Dear Dr Campellone,

We are pleased to inform you that your manuscript entitled "Genomic instability caused by Arp2/3 complex inactivation results in micronucleus biogenesis and cellular senescence" has been editorially accepted for publication in PLOS Genetics. Congratulations!

Yours sincerely,

Gregory P. Copenhaver

Editor-in-Chief

PLOS Genetics

Gregory Barsh

Editor-in-Chief

PLOS Genetics

Comments from the reviewers (if applicable):

**Data Deposition**

http://datadryad.org/submit?journalID=pgenetics&manu=PGENETICS-D-22-00090R2

**Press Queries**

---

## [Editor Report · Acceptance letter]

23 Jan 2023

PGENETICS-D-22-00090R2 

Genomic instability caused by Arp2/3 complex inactivation results in micronucleus biogenesis and cellular senescence 

Dear Dr Campellone, 

We are pleased to inform you that your manuscript entitled "Genomic instability caused by Arp2/3 complex inactivation results in micronucleus biogenesis and cellular senescence" has been formally accepted for publication in PLOS Genetics! Your manuscript is now with our production department and you will be notified of the publication date in due course.

With kind regards,

Zsofia Freund

PLOS Genetics

On behalf of:
